# Diffusion Models are Secretly Exchangeable: Parallelizing DDPMs via Autospeculation

Hengyuan Hu [* 1]   Aniket Das [* 1]   Dorsa Sadigh [1]   Nima Anari [1]

## Abstract

Denoising Diffusion Probabilistic Models (DDPMs) have emerged as powerful tools for generative modeling. However, their sequential computation requirements lead to significant inference-time bottlenecks. In this work, we utilize the connection between DDPMs and Stochastic Localization to prove that, under an appropriate reparametrization, the increments of DDPM satisfy an exchangeability property. This general insight enables near-black-box adaptation of various performance optimization techniques from autoregressive models to the diffusion setting. To demonstrate this, we introduce *Autospeculative Decoding* (ASD), an extension of the widely used speculative decoding algorithm to DDPMs that does not require any auxiliary draft models. Our theoretical analysis shows that ASD achieves a $\widetilde{O}(K^{1/3})$ parallel runtime speedup over the $K$ step sequential DDPM. We also demonstrate that a practical implementation of autospeculative decoding accelerates DDPM inference significantly in various domains.

## 1. Introduction

Diffusion models have emerged as one of the leading tools in generative modeling (Sohl-Dickstein et al., 2015; Ho et al., 2020; Song et al., 2021b). They are widely used to generate samples in continuous spaces, such as when an image (Dhariwal & Nichol, 2021; Nichol et al., 2021; Kingma et al., 2021; Rombach et al., 2022) or a sequence of continuous actions (Reuss et al., 2023; Chi et al., 2023) is desired. A major limitation of diffusion models, particularly limiting in real-time applications such as continuous control

*Equal contribution   [1]Computer Science Department, Stanford University, California, USA. Correspondence to: Hengyuan Hu <hengyuan.hhu@gmail.com>, Aniket Das <aniketd@stanford.edu>.

*Proceedings of the 42nd International Conference on Machine Learning*, Vancouver, Canada. PMLR 267, 2025. Copyright 2025 by the author(s).

or robotics, is their slow inference time. Standard implementations of Denoising Diffusion Probabilistic Models (DDPMs) take a large number of denoising steps, $\approx 1000$ steps in image models (Ho et al., 2020; Rombach et al., 2022), and $\approx 100$ steps in robotics models (Chi et al., 2023), to generate samples. Using fewer steps empirically results in a loss of quality; theoretical analysis of DDPMs suggests that $\approx \widetilde{O}(d)$ steps are needed for high-fidelity samples from $d$-dimensional distributions (Benton et al., 2023).

Numerous works have attempted to address slow inference. Works like DDIM (Song et al., 2021a) and DPMSolver (Lu et al., 2022) have lowered the number of steps by altering the inference method to a deterministic one, barring the initialization, and utilizing ODE solvers. Other works (Meng et al., 2022; Song et al., 2023; Watson et al., 2022; Lu & Song, 2024) have changed the underlying model, requiring new training, to enable fewer steps during inference. These speedups are all achieved by trading off the quality of the generated samples. Recently, Shih et al. (2024) showed that instead of trading off quality for speed, one can trade compute for speed by leveraging parallelization (e.g., utilizing multiple GPUs). However, the proposed method in (Shih et al., 2024) still leaves a small but tunable error as it utilizes a fixed point iteration method with early stopping.

Our work also focuses on accelerating inference in DDPMs via parallelization—*albeit without changing the quality of the generated samples.* Surprisingly, we show that it is possible to produce samples *identically distributed* as the sequential process, i.e., thereby ensuring *zero* quality loss, while also achieving a *theoretically guaranteed* speedup over sequential sampling. We prove that our method needs $\approx \widetilde{O}(d^{2/3})$ parallel calls to the model, much smaller than $\approx \widetilde{O}(d)$ in sequential sampling (Benton et al., 2023).

Our algorithm adapts speculative decoding, an acceleration technique for autoregressive models such as large language models (LLMs), to the setting of diffusion models. Traditionally speculative decoding uses an *additional* smaller LLM (or model), called the draft, to accelerate sampling from the main LLM. The draft model produces tokens that are then in *parallel* verified by the main model – leading to possible parallel speedup. At first sight, adapting speculative decoding to diffusion models seems challenging for

two reasons: first, the space is continuous; and second, a draft diffusion model is likely to follow very different trajectories from the main diffusion model, unlike two LLMs which can plausibly predict a few similar tokens at a time. We overcome these challenges by *eschewing a draft model altogether*, and instead show how to use the main diffusion model itself as its own draft. Our main insight enabling this *autospeculation* behavior is that the distribution of denoising trajectories in a diffusion model exhibits certain time symmetries. In technical terms, we show that after a syntactic time/scale change, the increments $\mathbf{y}_{t+\mathrm{d}t} - \mathbf{y}_t$ of a denoising trajectory $(\mathbf{y}_t)_{t \in [0,T]}$ form an exchangeable sequence of random variables – in other words, their distribution remains unchanged under any permutation. This allows us to use independent samples from the distribution of the next denoising step as speculations for all future denoising steps.

We emphasize that this secret property of diffusion models, which we refer to as *hidden exchangeability* is surprising. For example, in LLMs, a similar property would only hold if the sequences of tokens in the underlying distribution are permutation-invariant; that is the probability of "hello world" in the language is the same as "world hello." In addition to enabling autospeculation, we use exchangeability to derive our theoretically guaranteed $\approx \widetilde{O}(d^{1/3})$ speedup.

Our contributions can be summarized as follows:

**Hidden Exchangeability in DDPMs:** We leverage the equivalence between DDPMs and Stochastic Localization (Eldan, 2013; Chen & Eldan, 2022), recently established by Montanari (2023) to uncover a fundamental *hidden exchangeability property* for the trajectories of DDPM. In particular, we show that, after an appropriate transformation, the increments of the DDPM process are exchangeable, i.e., their joint law is permutation-invariant. This insight enables us to view DDPMs through the lens of any-order autoregressive models (Shih et al., 2022), potentially allowing the adaptation of performance optimization techniques previously limited to traditional autoregressive architectures.

**Autospeculative Decoding for DDPMs:** We use the hidden exchangeability property to design an efficient inference algorithm, which, at any given timestep $a$, makes a single call to the DDPM model to predict future increments, and subsequently makes *calls to the same model, all in one parallel step* to verify these predictions via rejection sampling. Our algorithm extends the Speculative Decoding (Leviathan et al., 2023) paradigm, widely used for LLM inference, to the diffusion model setting. Unlike traditional speculative decoding, our algorithm eschews an auxiliary draft model and instead leverages the hidden exchangeability property to make the original DDPM speculate about itself. Hence, we call our algorithm *Autospeculative Decoding (ASD)* and show that it performs error-free DDPM inference with massive parallelization.

**Theoretical Guarantees:** We prove that ASD is an *error-free parallelization algorithm*, whose output is *identically distributed* as sequential samples from the DDPM. Furthermore, under a minimal second-moment assumption, we prove that ASD makes at most $\widetilde{O}(d^{2/3})$ parallel calls to the model on $d$-dimensional distributions, as compared to $\widetilde{O}(d)$ calls needed in the sequential implementation of DDPM (Benton et al., 2023). To the best of our knowledge, our result is the *first* parallel inference algorithm for DDPMs that shows *empirical speedups* and has a *theoretically guaranteed speedup* without any restrictive assumptions on the score function such as Lipschitz continuity.

**Empirical Evaluation:** We complement our theoretical contributions with extensive empirical evaluations on diffusion models for image generation and robot control tasks. ASD leads to 1.8-4$\times$ speedup in wall-clock time without any loss in quality.

### 1.1. Notation

We analyze diffusion models defined on Euclidean spaces $\mathbb{R}^d$. We use $\mathbf{x}, \mathbf{y}$ to represent vectors and $\mathbf{M}$ to represent matrices. $\mathbf{I}$ is the identity matrix. For a random variable $\mathbf{z}$, we use $\mathsf{Law}(\mathbf{z})$ to denote its distribution. For random variables $\mathbf{x}$ and $\mathbf{y}$, we use $\mathbf{x} \overset{d}{=} \mathbf{y}$ to denote $\mathsf{Law}(\mathbf{x}) = \mathsf{Law}(\mathbf{y})$. For any measure $\mu$, $\mathsf{Cov}[\mu]$ denotes its covariance. We use $\mathsf{TV}\left(.,.\right)$ and $\mathsf{KL}\left(.\|.\right)$ to denote the Total Variation distance and KL divergence. $B_t$ and $W_t$ denote standard Brownian motions on $\mathbb{R}^d$ unless stated otherwise. We use the $O, \Omega, \Theta$ notation to suppress dependence on numerical and problem specific constants and $\tilde{O}, \tilde{\Omega}, \tilde{\Theta}$ to suppress logarithmic factors. $\lesssim, \gtrsim$ and $\asymp$ denote $\leq, \geq$ and $=$ modulo universal constants.

## 2. Related Work

Several prior works, notably Shih et al. (2024) and Pokle et al. (2022), have shown how to use parallelization to empirically accelerate sampling from diffusion models. These works use a fixed point iteration, also called the Picard iteration or the collocation method, to break the sequential nature of denoising steps in diffusion models. These methods use heuristics to stop the fixed point iteration when approximate convergence is detected, and thus leave a small error in the samples, unlike our results. Later works of Gupta et al. (2024); Chen et al. (2024) provided theoretical convergence guarantees for these parallelization techniques, but under the very restrictive assumption that the score functions of the underlying distribution and all of its evolutions under the forward process of DDPM satisfy $L$-Lipschitzness for a very small $L$. In particular, when the underlying distribution is not smooth, or even when $L$ is a small polynomial of the dimension $d$, the guarantees become vacuous.

The work of Benton et al. (2023) provides the best-known theoretical analysis for the runtime of diffusion models in the sequential setting, namely $\widetilde{O}(d)$ denoising steps for $d$-dimensional distributions. Without restrictive assumptions on the underlying distribution, such as Lipschitzness of scores, this $\widetilde{O}(d)$ guarantee remained the best-known theoretical result even in the parallel setting. Our work provides a parallel speedup under the minimal assumption of boundedness of second moments, essentially the same assumption as in the work of Benton et al. (2023).

Our work adapts speculative decoding (Leviathan et al., 2023; Chen et al., 2023), a parallelization technique widely used for LLMs and autoregressive models, to the setting of diffusion models. In the special setting of any-order autoregressive models (Shih et al., 2022), which excludes most LLMs, Anari et al. (2024a) provided the first theoretically guaranteed speedup for speculative decoding. This, combined with our insight on the exchangeability of diffusion models, was the main source of inspiration for our work. Anari et al. (2024a) showed a speedup of $\widetilde{O}(d^{1/3}/\text{poly}\log(q))$ for generating a sequence of $d$ tokens, if the token space is of size $q$. We note that, while our proof adopts many elements from the work of Anari et al. (2024a), there are significant challenges that we overcome: first, diffusion models live in a continuous space, roughly speaking this corresponds to $q = \infty$, for which the speedup guarantees of Anari et al. (2024a) become meaningless; and second, in diffusion models, even the sequential sampling process is a discretization of a Stochastic Differential Equation (SDE), and the error resulting from the discretization makes the analysis particularly challenging.

The concurrent and independent work of De Bortoli et al. (2025), which came to our notice post-submission, also proposes the idea of parallelizing diffusion models via speculative decoding. In particular, their idea of a Frozen Target Draft Model corresponds to Autospeculative Decoding. To the best of our understanding, De Bortoli et al. (2025) does not establish any parallel speedup guarantees for their algorithm, and their algorithm design is not based on the hidden exchangeability property.

## 3. Exchangeability in Diffusion Models

We first present a brief introduction to DDPMs and refer the readers to Song et al. (2021b); Chen et al. (2022) for a broader discussion. For an unknown target distribution $\mu$ on $\mathbb{R}^d$, we consider the following forward or noising process $\bar{\mathbf{x}}_t^{\rightarrow}$ for time $t \in [0, T]$.

$$\mathrm{d}\bar{\mathbf{x}}_t^{\rightarrow} = h(t)\bar{\mathbf{x}}_t^{\rightarrow}\,\mathrm{d}t + \sqrt{u(t)}\,\mathrm{d}W_t, \quad \bar{\mathbf{x}}_0^{\rightarrow} \sim \mu \quad (1)$$

where $h(t), u(t)$ are arbitrary continuous functions with $u(t) > 0$. Notable special cases include the Variance Preserving SDE (VP-SDE) which sets $h(t) = -\frac{1}{2}u(t)$ and the

Variance Exploding SDE (VE-SDE) which sets $h(t) = 0$. Under mild conditions, $\mu_t = \mathsf{Law}(\bar{\mathbf{x}}_t^{\rightarrow})$ converges to a tractable distribution which is easy to sample from. For instance, the choice $u(t) = 2$ and $h(t) = -1$ corresponds to the Ornstein Uhlenbeck (OU) SDE, which converges to a standard Guassian at an exponential rate (Bakry et al., 2014), and hence, satisfies $\mu_T \approx \mathcal{N}(0, \mathbf{I})$.

The reverse process, i.e., the denoising process $\bar{\mathbf{x}}_t^{\leftarrow} \sim \mu_{T-t}$ is governed by the following SDE

$$\mathrm{d}\bar{\mathbf{x}}_t^{\leftarrow} = f(t, \bar{\mathbf{x}}_t^{\leftarrow})\,\mathrm{d}t + \sqrt{u(T-t)}\,\mathrm{d}W_t$$
$$f(t, \bar{\mathbf{x}}_t^{\leftarrow}) = -h(T-t)\bar{\mathbf{x}}_t^{\leftarrow} + u(T-t)\nabla \ln \mu_{T-t}(\bar{\mathbf{x}}_t^{\leftarrow})$$
$$(2)$$

The working principle behind DDPMs is as follows: Since $\mu_T$ is easy to approximately sample from (e.g $\mu_T \approx \mathcal{N}(0, \mathbf{I})$ for the OU DDPM), access to an (approximate) oracle for the score function $\nabla \ln \mu_{T-t}(\bar{\mathbf{x}}_t^{\leftarrow})$ gives us a generative model for $\mu$ which involves sampling $\mathbf{y}_0 \sim \nu$ (where $\nu \approx \mu_T$) and then following the dynamics of (2). Algorithmically, this is implemented via the following Euler discretization:

$$\mathbf{y}_{i+1} = \mathbf{y}_i + \eta_i f(t_i, \mathbf{y}_i) + \sigma_{i+1}\xi_{i+1}, \quad \xi_{i+1} \sim \mathcal{N}(0, \mathbf{I})$$
$$(3)$$

Where $t_0 \leq t_1 \leq \ldots t_{K-1}$ are discretization points, $\eta_i = t_{i+1} - t_i$ is the step-size and $\sigma_{i+1} = \sqrt{\eta_i g(T - t_i)}$.

### 3.1. Hidden Exchangeability of DDPMs

We now introduce Stochastic Localization (SL), an analytic tool developed by Eldan (2013), which has led to several breakthroughs in probability and theoretical computer science (Chen, 2021; El Alaoui et al., 2022; Benton et al., 2023; Anari et al., 2024b). Given a target distribution $\mu$ on $\mathbb{R}^d$, SL is defined by the following process $\bar{\mathbf{y}}_t$:

$$\bar{\mathbf{y}}_t = \mathbf{m}(t, \bar{\mathbf{y}}_t)\,\mathrm{d}t + \mathrm{d}B_t, \quad \bar{\mathbf{y}}_0 = 0$$
$$\mathbf{m}(t, \mathbf{y}) = \mathbb{E}_{\mathbf{x}^\star \sim \mu, \, \xi \sim \mathcal{N}(0, \mathbf{I})}[\mathbf{x}^\star \mid t\mathbf{x}^\star + \sqrt{t}\xi = \mathbf{y}] \quad (4)$$

It is known that $\mathsf{Law}(\bar{\mathbf{y}}_t/t) = \mu * \mathcal{N}(0, \mathbf{I}/t)$ and in particular as $t \to \infty$, we have $\mathsf{Law}(\bar{\mathbf{y}}_t/t) \to \mu$ (El Alaoui & Montanari, 2022). To this end, stochastic localization resembles the reverse process in DDPMs as it can also function as a generative model for $\mu$ given oracle access to $\mathbf{m}(t, \mathbf{y})$. This similarity was demystified by Montanari (2023), who proved that the OU DDPM is equivalent to the SL process. In Appendix B, we extend their result to show that arbitrary DDPMs are equivalent reparametrizations of Stochastic Localization. In particular, we prove the following

**Theorem 1** (Equivalence of DDPM and SL). *Let $\bar{\mathbf{x}}_t^{\leftarrow}$ denote the DDPM reverse process defined in (2) and let $\bar{\mathbf{y}}_t$ denote the SL process defined in (4). Then, there exist continuous invertible functions $\gamma(t), \zeta(t)$ that are uniquely determined by $h(t)$ and $u(t)$ such that $\bar{\mathbf{y}}_t \stackrel{d}{=} \gamma(t)\bar{\mathbf{x}}_{\zeta(t)}^{\leftarrow}$*

The key insight behind our algorithm comes from the following time-invariance property of SL. We present a proof of this result and a detailed discussion on the properties of SL in Appendix B.

**Theorem 2** (Exchangeability in SL increments). *Let $\bar{\mathbf{y}}_t$ denote the SL process defined above. Consider any $t_1 \leq t_2 \leq \ldots \leq t_m$ and let $\eta_i = t_{i+1} - t_i$. Then, the increments of the SL process satisfies the following time-invariance property for any $\pi \in \mathbb{S}_{m-1}$:*

$$\mathsf{Law}((\mathbf{y}_{t_i+\eta_i} - \mathbf{y}_{t_i})_{i\in[m-1]}) = \mathsf{Law}((\mathbf{y}_{t_{\pi(i)}+\eta_i} - \mathbf{y}_{t_{\pi(i)}})_{i\in[m-1]})$$

*In particular, if the time increments $\eta_i$ are all equal, then the increments $\Delta_i = \mathbf{y}_{t_{i+1}} - \mathbf{y}_{t_i}$ are exchangeable, i.e., $\mathsf{Law}((\Delta_i)_{i\in[m-1]}) = \mathsf{Law}((\Delta_{\pi(i)})_{i\in[m-1]}) \ \forall \ \pi \in \mathbb{S}_{m-1}$*

Since DDPMs and SL are equivalent reparametrizations of each other, we refer to Theorem 2 as the *hidden exchangeability* property of DDPMs. In essence, Theorem 2 allows us to use the DDPM increments at any given timestep as a proposal distribution for sampling the increments of future timesteps. As we shall demonstrate, this proposal and its subsequent verification can each be performed in parallel, leading to a highly efficient inference algorithm.

Note that, assuming a fine discretization, calls to the function $\mathbf{m}(\cdot)$, equivalent to the trained model in DDPMs, precisely allow us to find the conditional distributions of the increments: $\mathsf{Law}(\Delta_i \mid \Delta_{<i})$. So in a way, one can view diffusion models as autoregressive models for the increments, with the major caveat that the token space here, $\mathbb{R}^d$, is infinite. Theorem 2 shows that $\mathsf{Law}(\Delta_j \mid \Delta_{<i})$ is the same for all $j \geq i$. So in particular at any point in the denoising process we can produce the marginal distribution of all future increments – it is the same as the immediately next increment. Independent samples from these marginals form the proposal/speculation in our algorithm, as was done by Anari et al. (2024a) for any-order autoregressive models.

## 4. Autospeculative Decoding

In this section, we describe AutoSpeculative Decoding (ASD), our main algorithmic contribution. We consider the general problem of sampling from Euler discretizations of SDEs of the following form:

$$\mathbf{y}_{i+1} = \mathbf{y}_i + \eta_i g(t_i, \mathbf{y}_i) + \sigma_{i+1}\xi_{i+1}, \ \xi_{i+1} \sim \mathcal{N}(0, \mathbf{I}) \tag{5}$$

where $t_0 \leq \ldots \leq t_k$ denotes a sequence of time-steps and $\eta_i = t_{i+1} - \eta_i$ denotes the step-size. We note that it this includes the Euler discretization of SL as a special case. Although typical discretizations of the DDPM don't directly fit into equation (5), this can be easily remedied by first translating the DDPM iterate to an SL iterate via the reparametrization discussed in Section 3, incrementing by

one time-step in the SL formulation and then mapping it back to the DDPM formalism.

At any iteration $i$, the conditional distribution of the next iterate $q(\mathbf{y}_{i+1}|\mathbf{y}_i)$, which we call the target distribution at step $i$, is of the following form:

$$q(\mathbf{y}_{i+1}|\mathbf{y}_i) = \mathcal{N}(\mathbf{y}_{i+1} \mid b(\eta_i, \mathbf{y}_i), \sigma_i)$$
$$b(\eta_i, \mathbf{y}_i) = \mathbf{y}_i + \eta_i g(t_i, y_i) \tag{6}$$

We refer to $b(\eta_i, \mathbf{y}_i)$ as the *target mean at step $i$*. Computing $b$ involves calling an (approximate) oracle for $g$, typically implemented via a neural network. Oracle calls to $g$, which we refer to as model calls, represent the primary computational bottleneck to sampling from equation Eq. (5). Note that for the SL process, $g(\cdot)$ is the same as the mean-predicting function $\mathbf{m}(\cdot)$ from Eq. (4).

To leverage the hidden exchangeability property of DDPMs, our algorithm does the following: At any given step $a$, it tries to predict or *speculate* the target mean of the future timesteps by sampling from a proposal distribution $p$, defined as follows. The proposal distribution is explicitly designed such that speculating the mean of future timesteps requires only one call to $g$ (i.e., one model call).

$$p\left((\mathbf{y}_{i+1})_{i\geq a}|\mathbf{y}_a\right) = \prod_{i\geq a} p(\mathbf{y}_{i+1}|\mathbf{y}_i, \mathbf{y}_a)$$
$$p(\mathbf{y}_{i+1}|\mathbf{y}_i, \mathbf{y}_a) = \mathcal{N}(\mathbf{y}_{i+1}|\hat{b}(\eta_i, \mathbf{y}_i, \mathbf{y}_a), \sigma_i)$$
$$\hat{b}(\eta_i, \mathbf{y}_i, \mathbf{y}_a) = \mathbf{y}_i + \eta_i g(t_a, \mathbf{y}_a) \tag{7}$$

We call $\hat{b}(\eta_i, \mathbf{y}_i, \mathbf{y}_a)$ the *proposal mean for step $i$ at step $a$*, and note that it requires only one call to $g$. In fact, it can even be computed in $\tilde{O}(1)$ parallel time via prefix sums.

Equipped with the above definitions, we present Autospeculative Decoding in Algorithm 1. The algorithm, which resembles Speculative Decoding (Leviathan et al., 2023; Chen et al., 2024) has three key steps: 1) Sampling from the proposal distribution via one model call, 2) Speculating the means of the future target distributions via the proposal samples, 3) Verifying the accuracy of the speculations via rejection sampling and resampling at the first disagreement. The verification procedure, stated in Algorithm 2 uses Gaussian Rejection Sampler, Algorithm 3, to simultaneously sample from the conditional target distribution and check if the proposal agrees with the target. Algorithm 3 is motivated by the reflection coupling technique of Bou-Rabee et al. (2020) and runs in $O(1)$ time by leveraging the fact that the proposal and target are Gaussians with the same variance.

*Remark 3.* Although the speculated means for the DDPM process can be derived by following the recipe described before for SL and using the equivalence of SL and DDPM, the end result has an intuitive form. One can rewrite the

---

**Algorithm 1:** Autospeculative Decoding (ASD)

**Input:** Steps $K$, Step-sizes $(\eta_k)_{k<K}$, Variances $(\sigma_k^2)_{k\in[K]}$, Initial $\mathbf{y}_0$, Speculation length $\theta$

1   sample $(u_1,\ldots,u_K) \sim \text{Uniform}([0,1]^K)$
2   sample $(\xi_1,\ldots,\xi_K) \overset{i.i.d.}{\sim} \mathcal{N}(0,\mathbf{I})$
3   $a \leftarrow 0$
4   **while** $a < K$ **do**
5      $\hat{\mathbf{y}}_a \leftarrow \mathbf{y}_a$, $b \leftarrow \min(K, a+\theta)$
      // compute proposal means and
        proposal samples
6      $\mathbf{v}_a \leftarrow g(t_a, \mathbf{y}_a)$
7      **for** $i = a,\ldots,b-1$ **do**
8        $\hat{\mathbf{m}}_{i+1} \leftarrow \hat{\mathbf{y}}_i + \eta_i \mathbf{v}_a$
9        $\hat{\mathbf{y}}_{i+1} \leftarrow \hat{\mathbf{m}}_{i+1} + \sigma_{i+1}\xi_{i+1}$
      // speculate target means in
       parallel
10     **for** $i = a,\ldots,b-1$ **in parallel**
11       $\mathbf{m}_{i+1} \leftarrow \hat{\mathbf{y}}_i + \eta_i g(t_i, \mathbf{y}_i)$
      // verify speculations via
       rejection sampling
12     $[(\mathbf{z}_i)_{a<i\le b}, j] \leftarrow \text{Verifier}((u_i,\xi_i,\hat{\mathbf{m}}_i,\mathbf{m}_i)_{a<i\le b})$
      // advance until first rejection
13     **if** $j < b$ **then**
14       $\mathbf{y}_i \leftarrow \mathbf{z}_i \ \forall i \in [a+1, j+1]$
15       $a \leftarrow j+1$
16     **else**
17       $\mathbf{y}_i \leftarrow \mathbf{z}_i \ \forall i \in [a+1, j]$
18       $a \leftarrow j$
19   **return** $(\mathbf{y}_0,\ldots,\mathbf{y}_K)$

---

DDPM process (3) by expanding out $f$ to get the form

$$\mathbf{y}_{i+1} = \alpha_i \mathbf{y}_i + \beta_i \mathbb{E}[\bar{\mathbf{x}}_0^\rightarrow \mid \mathbf{y}_i] + \sqrt{\eta_i}\xi_{i+1}$$

for some coefficients $\alpha_i, \beta_i$. In many implementations of DDPM, the model is trained to output this $\mathbb{E}[\bar{\mathbf{x}}_0^\rightarrow \mid \mathbf{y}_i]$ directly instead of $f$. At step $a$, to speculate the means of future steps $i$, we simply need to plugin $\mathbb{E}[\bar{\mathbf{x}}_0^\rightarrow \mid \mathbf{y}_a]$ for $\mathbb{E}[\bar{\mathbf{x}}_0^\rightarrow \mid \mathbf{y}_i]$ in the above formula.

**Parallelization in Algorithm 1** In each iteration, ASD makes one model call in line 6 to compute the proposal means and samples, and one *parallel* round of model calls in line 11 to speculate the target means. Beyond this, the remaining internal computation of ASD is also highly parallelizable. In particular, each for loop in Algorithm 1 and Algorithm 2 is parallelizable and Algorithm 3 takes $O(1)$ time.

---

**Algorithm 2:** Verifier

**Input:** Uniform Random Seeds $(u_{a+1},\ldots,u_b)$, Speculated Means $(\hat{\mathbf{m}}_{a+1},\ldots,\hat{\mathbf{m}}_b)$, Target Means $(\mathbf{m}_{a+1},\ldots,\mathbf{m}_b)$, Variance Schedule $(\sigma_{a+1}^2,\ldots,\sigma_b^2)$

1   $j \leftarrow a+1$
2   **for** $i = a+1,\ldots,b$ **in parallel**
      // GRS defined separately in
       Algorithm 3
3     $(\mathbf{z}_i, b_i) \leftarrow \text{GRS}(u_i, \xi_i, \hat{\mathbf{m}}_i, \mathbf{m}_i, \sigma_i^2)$
4     **if** $b_i = True$ and $j < i$ **then**
5       $j \leftarrow i$
6   **return** $[(\mathbf{z}_{a+1},\ldots,\mathbf{z}_b), j]$

---

**Algorithm 3:** Gaussian Rejection Sampler (GRS)

**Input:** $u \sim \text{Uniform}([0,1])$, $\xi \sim \mathcal{N}(0,\mathbf{I})$, Proposal mean $\hat{\mathbf{m}}$, Target mean $\mathbf{m}$, Variance $\sigma^2$

1   $\mathbf{v} \leftarrow \hat{\mathbf{m}} - \mathbf{m}$
2   $b \leftarrow \mathbb{1}\left[u \le \min\left(1, \frac{\mathcal{N}(\xi+\sigma^{-1}\mathbf{v}|0,\mathbf{I})}{\mathcal{N}(\xi|0,\mathbf{I})}\right)\right]$
3   **if** $b = True$ **then**
4     $\mathbf{x} \leftarrow \hat{\mathbf{m}} + \sigma\xi$
5   **else**
6     $\mathbf{x} \leftarrow \mathbf{m} + \sigma\left(\xi - 2\mathbf{v}\cdot\frac{\langle\mathbf{v},\xi\rangle}{\|\mathbf{v}\|^2}\right)$
7   **return** $(\mathbf{x}, b)$

---

## 5. Theoretical Guarantees

In this section, we present our theoretical guarantees for Algorithm 1. Our first result, which is proved in Appendix C.2, guarantees that ASD is an *error-free parallelization method*, i.e. it always samples exactly from the target distribution.

**Theorem 4** (Correctness of Algorithm 1). *Algorithm 1 terminates in at most $K$ steps and its outputs are always distributed exactly according to the target distribution of the stochastic process* (5).

We now analyze the parallel runtime of Algorithm 1 for the Euler discretization of the SL process, and prove the following bound on the adaptive complexity, i.e., the number of parallel model calls, in Appendix C.3. While our proof is specific to the SL process, our result implies a similar guarantee for DDPMs due to its equivalence to SL, as discussed in Section 3.1

**Theorem 5** (Adaptive Complexity of Algorithm 1). *Suppose the data distribution satisfies $\text{Tr}(\text{Cov}[\mu]) \le \beta d$ and the step-sizes of the DDPM satisfy $\eta_k \le \eta$. Then, for $\theta \asymp (K/\beta\eta d)^{1/3}$, Algorithm 1 makes at most $O(K^{2/3}(\beta d\eta)^{1/3})$ parallel calls to the DDPM in expectation. In fact, for any $\delta \in (0,1)$, the number of parallel calls to the DDPM is*

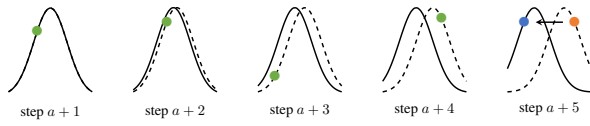

Figure 1: A depiction of AutoSpeculative Decoding. Speculated Gaussians with proposal means $\hat{\mathbf{m}}_i$ are shown in dashed and Gaussians with target means $\mathbf{m}_i$ are shown in solid. Green and orange points are proposed samples $\hat{\mathbf{y}}_i$. The verifier accepts the first four proposals, rejects the fifth, and replaces the fifth proposed sample with its reflection. Proposed samples for steps beyond the fifth are ignored.

bounded by $O(K^{2/3}(\beta d\eta)^{1/3}\log(1/\delta))$ *with probability at least* $1-\delta$.

$\tilde{O}(K^{1/3})$ **Parallel Speedup:** In accordance with prior works analyzing DDPMs, $\eta d = O(1)$ is usually required to ensure convergence to the target distribution (Chen et al., 2022; Benton et al., 2023). Moreover, $\beta$ is typically $O(1)$. Under this setting, the number of parallel DDPM calls made by Algorithm 1 is $O(K^{2/3})$ which represents a $O(K^{1/3})$ parallel speedup over the vanilla sequential DDPM.

**Comparison to Prior Works:** The work of (Shih et al., 2024) proposed a parallel algorithm for DDPMs based on Picard iterations but did not provide any theoretical guarantees. Along similar lines, the works of (Gupta et al., 2024; Chen et al., 2024) combined Picard iterations with the Randomized Midpoint Method to design a parallel algorithm for DDPMs with $O(\text{polylog}(d))$ parallel runtime, assuming bounded second moments and $O(1)$ uniform Lipschitzness of the score function, a stringent assumption which is satisfied for very restricted distribution families. As discussed before, these approaches are *not* error-free parallelization schemes due to the use of Picard iterations. On the contrary, ASD is a perfect sampler with guaranteed speedups even in the absence of any smoothness assumptions on the score function.

**From Adaptive Complexity to Parallel Runtime:** While Theorem 5 upper bounds the number of parallel calls to the DDPM, we note that the parallel runtime (i.e., time taken on a PRAM) of the Algorithm satisfies the same guarantee modulo logarithmic factors. This is because the internal computation of Algorithm 1 is easily parallelizable. For instance, the proposal and the target means can be computed in $\tilde{O}(1)$ parallel time via prefix sums, and the for loop in Algorithm 2 is also parallelizable.

## 6. Experiments

In the experiments, we empirically demonstrate the practical benefits of autospeculative decoding (ASD). We focus on two key properties of ASD: 1) accelerating inference

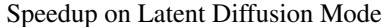

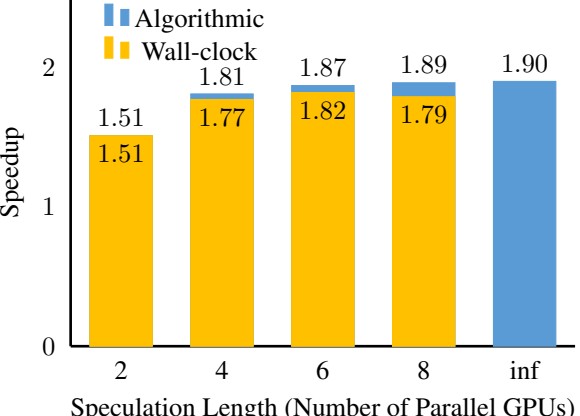

Figure 2: Speedup of ASD over DDPM on StableDiffusion-v2 with different speculation length $\theta$. The *algorithmic* speedup measures the reduction in the number of calls to the noise prediction network while *wall-clock* speedup measures the reduction in wall-clock time of the denoising process by running ASD.

through multi-step speculation and parallel verification and 2) producing truthful samples of the underlying diffusion models. We consider a diverse set of real-world applications where diffusion models are used, including image generation with both latent- and pixel-space diffusion models (Rombach et al., 2022; Ho et al., 2020) and robot control with diffusion policies (Reuss et al., 2023; Chi et al., 2023). ASD offers 2-7× reduction in the number of sequential neural network predictions via speculation and our practical implementation achieves 1.8-4× speedup measured in wall-clock time. We emphasize that our goal is *not* to claim any state-of-the-art acceleration results for diffusion models but to provide empirical evaluations complementary to our theoretical contributions that enables fast, exact sampling without approximation.

To measure the speedup of ASD over vanilla DDPM, we consider two scenarios referred to as ***algorithmic*** and ***wall-clock*** throughout the paper. For the *algorithmic* speedup, we divide the total number of denoising steps (e.g. 1000 in DDPM for image generation and 100 in diffusion policy for robot control) by the number of times ASD invokes the internal neural network to make predictions. This mainly helps us understand the rate at which ASD progresses. It also corresponds to the ideal speedup assuming perfect parallelization where parallel predictions take the same amount of time as a single prediction, and ignoring the minor overheads of other non-neural network computations such as speculation and rejection sampling. The *wall-clock* speedup measures the reduction in the wall-clock time of full denois-

| DDPM | ASD-2 | ASD-4 | ASD-6 | ASD-8 | ASD-$\infty$ |
|------|-------|-------|-------|-------|--------------|
| 32.1 | 32.2 | 32.0 | 32.2 | 32.3 | 32.1 |

Table 1: **CLIP score** (the higher the better) of images generated from DDPM and ASD-$\theta$ with different speculation length $\theta$s. The CLIP scores are computed over 1000 captions from the COCO2017 captions validation dataset. ASD does not affect the quality of generated images.

| DDPM | ASD-4 | ASD-6 | ASD-8 | ASD-$\infty$ |
|------|-------|-------|-------|--------------|
| 14.3 | 13.2 | 13.2 | 13.2 | 14.3 |

Table 2: **FID score** (the lower the better) of images sampled from DDPM and ASD with different speculation length. The underlying diffusion model is trained on the LSUN Church dataset. Each score is computed with 5000 image samples. Samples from ASD have the same quality as the ones from non-speculative DDPM.

ing loops, accounting for all extra overheads such as data transferring cost when parallelizing over multiple GPUs or the additional time for batched prediction.

### 6.1. Accelerating Diffusion for Image Generation

**Latent Diffusion Models:** We first evaluate ASD on image generation with latent diffusion models (LDM). Specifically, we use open-sourced StableDiffusion-v2 (Rombach et al., 2022; Schuhmann et al., 2022) model from the diffusers (von Platen et al., 2022) library. To implement ASD with parallelization, we first produce $\theta$ proposals following lines 8-9 in Algorithm 1 and then distribute the $\theta$ model prediction steps in line 11 over $\theta$ GPUs with multi-processing. Finally, we transfer the results back to the main process and run the acceptance and rejection sampling of lines 12-18. Similar to Shih et al. (2024), we choose to implement parallel prediction through multiple GPUs instead of batching because the network is large enough that the time it takes to make a prediction grows linearly with the batch size even when the batch is small. In cases where the network is smaller, such as in the robot control experiments discussed later, batching may still be a valid solution to achieve high acceleration without incurring additional hardware cost.

Fig. 2 shows the *algorithmic* and *wall-clock* speedup of ASD relative to DDPM under 1000 denoising steps. We evaluate ASD with different speculation length $\theta$ including infinity to understand the upper limit of ASD. As expected, longer speculation length in ASD leads to higher *algorithmic* speedup as it allows for more parallelism. ASD-$\infty$ achieves $1.9\times$ speedup, which corresponds to reducing the number of sequential predictions by nearly a factor of 2 when counting predictions that happen in parallel as 1. In practice, $\theta = 6$ or $\theta = 8$ is sufficient to achieve a similar

*algorithmic* speedup as ASD-$\infty$. We measure the *wall-clock* speedup on a machine with 8 NVIDIA A40 GPUs. Our implementation achieves a peak speedup of $1.82\times$ with $\theta = 6$. In our specific implementation and hardware setup, the overhead of creating and transferring data for more speculation steps outweighs the marginal benefit of bigger $\theta$.

To verify that ASD produces truthful samples from the underlying distribution, we compute the CLIP score (Hessel et al., 2021) using 5000 images generated with languages from COCO2017 captions validation dataset. Results in Table 1 show that the samples from ASD have the same quality as the ones generated by original DDPM. Fig. 3 shows the generated images from vanilla DDPM and ASD-$\infty$ using the same prompts side by side.

**Pixel Diffusion Models:** We also evaluate ASD on the LSUN Church model from Ho et al. (2020), which directly generates images of resolution $3 \times 256 \times 256$. We use the same ASD implementation as in the latent diffusion case with up to 8 GPUs for parallel computation. Fig. 4 shows the *algorithmic* and *wall-clock* speedup of ASD on this model. ASD achieves higher speedup here than in the previous latent diffusion model, reducing the number of neural network prediction calls up to $3.1\times$. However, we also notice that the gap between the *wall-clock* and *algorithmic* speedup is more significant in this case for two reasons. First, despite being a pixel space diffusion model, the computation cost per forward of this specific model is actually $50\%$ cheaper than the latent model from the previous section. Second, the overhead of transferring inputs and predictions between processes is roughly $10\times$ higher due to higher resolution and higher floating point precision. These two factors lead to a more prominent overhead and thus bigger gap between *wall-clock* and *algorithmic* speedup than in previous experiment. A faster implementation on systems with faster inter-connection between GPUs may further close the gap. The FID scores in Table 2, computed over 5,000 samples for each method, confirm that ASD produces samples of the same quality as DDPM.

### 6.2. Accelerating Diffusion for Robot Control

Diffusion models have become a popular method in learning robot control policies from demonstrations (Chi et al., 2023; Reuss et al., 2023). It models the conditional distribution of an action sequence $\pi(a_{t:t+k}|o_t)$ where $o_t$ is the observations from cameras attached to the robot. Conceptually, the process can be viewed as generating a small 2D vector of size $k \times d$ where $k$ is the length of the action sequence and $d$ is the action dimension. We consider three hard Robomimic (Mandlekar et al., 2021) simulation environments namely Square, Transport and Tool Hang. We follow prior works to set $k = 16$ in all environments. The action dimension $d = 7$ in the single arm Square and Tool

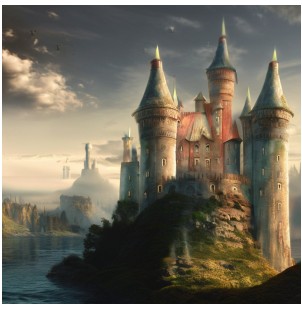 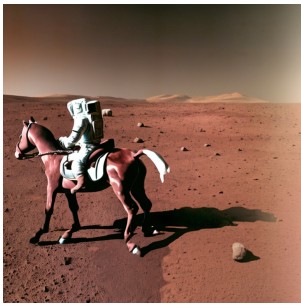 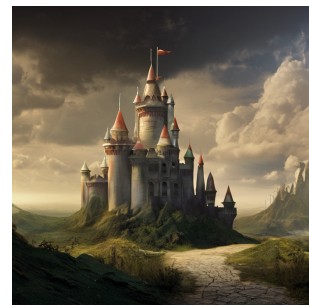 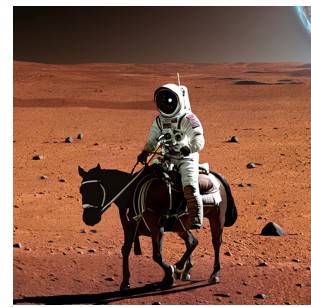

(a) DDPM: a beautiful castle, matte painting

(b) DDPM: a photo of an astronaut riding a horse on mars

(c) ASD-∞: a beautiful castle, matte painting

(d) ASD-∞: a photo of an astronaut riding a horse on mars

Figure 3: Image generated by StableDiffusion-v2 with different sampling method, DDPM and ASD-∞.

| Env | DDPM | ASD-8 | ASD-12 | ASD-16 | ASD-20 | ASD-24 | ASD-∞ |
|---|---|---|---|---|---|---|---|
| Square | $88.67 \pm 0.27$ | $93.00 \pm 1.70$ | $91.67 \pm 1.66$ | $93.00 \pm 0.47$ | $93.00 \pm 0.47$ | $90.00 \pm 1.89$ | $90.33 \pm 0.27$ |
| Transport | $89.00 \pm 1.25$ | $90.00 \pm 1.25$ | $91.33 \pm 1.52$ | $90.00 \pm 0.47$ | $91.00 \pm 0.00$ | $90.00 \pm 0.82$ | $90.00 \pm 1.25$ |
| Tool Hang | $70.00 \pm 2.94$ | $69.33 \pm 2.60$ | $66.00 \pm 2.62$ | $71.00 \pm 2.49$ | $70.00 \pm 2.36$ | $68.67 \pm 1.19$ | $73.67 \pm 0.27$ |

Table 3: Performance on Robomimic Tasks. ASD with different speculation length achieves similar performance as the vanilla DDPM. The number in each cell is obtained by evaluating the diffusion policy on 100 random seeds (random initial configurations of the scene) and repeating 3 times. We report the mean ± standard error of mean.

Hang tasks and $d = 14$ in the bi-manual Transport task.

The denoising neural network in the diffusion policies is considerably more lightweight than the ones from image diffusion models in the previous section. Therefore, we opt for a batching implementation for the parallel prediction step instead of parallelizing over multiple GPUs. We simply batch the proposal samples $\hat{\mathbf{y}}_i$ and call the network on the batch to predict the noise $\epsilon$ using *one GPU*.

Fig. 5 shows the speedup results of ASD on diffusion policies in all three tasks, relative to the vanilla DDPM that runs for 100 steps. Empirically, we find that ASD has a much higher acceptance rate for the speculated samples in these cases, leading to a 6-7× *algorithmic* speedup for ASD-∞. Due to the high acceptance rate, it requires a larger speculation length of 20 or 24 to match the efficiency of ASD-∞. The practical implementation achieves roughly 4× *wall-clock* speedup across the three tasks. The gap between *algorithmic* and *wall-clock* is noticeably larger than the gap in image generation experiments because the overhead for speculation and rejection sampling is more prominent given the cheaper compute cost of the neural network prediction. The extra cost of forwarding the network on batched input over a single input also contributes to the gap.

We unroll the sampled actions in each environment to verify that the quality of samples from ASD remains the same as the ones from original DDPM. Table 3 summarizes the results. In each environment, we evaluate the same diffusion policy with different sampling schemes over the same set of 100 seeds (100 random initial configurations) and repeat three times to account for other randomness in the evaluation process. ASD variants achieve a similar success rate as vanilla DDPM across all three environments.

# 7. Conclusion and Future Work

We introduce ASD, an error-free parallelization framework for DDPMs that achieves a guaranteed $\widetilde{O}(K^{1/3})$ parallel speedup under minimal assumptions and delivers 1.8-4× acceleration in practical domains. Our work establishes a fundamental bridge between efficient inference techniques for LLMs and diffusion models, enabling cross-pollination between these seemingly disjoint areas. Future directions include extending our framework to discrete diffusion models like SEDD (Lou et al.) and developing theoretical guarantees for speculative decoding in language models under realistic assumptions.

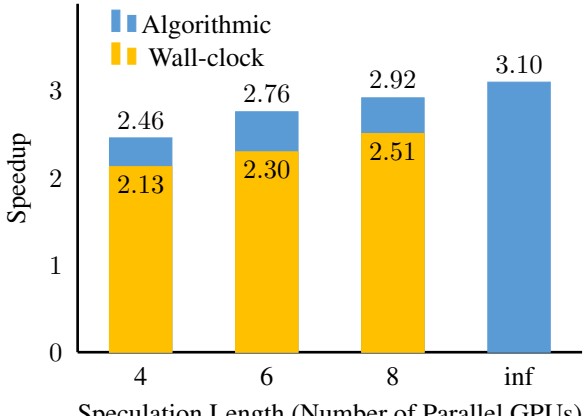

Figure 4: Speedup of ASD over DDPM using diffusion model from Ho et al. (2020). The model directly generates images with resolution $3 \times 256 \times 256$. ASD achieves up to $3.1 \times$ *algorithmic* speedup. However, compared to LDM, the overhead of data transfer between processes/GPUs is bigger while the computation cost is cheaper, leading to a wider gap between *wall-clock* and *algorithmic* speedup.

## Impact Statement

This paper presents work whose goal is to advance the field of Machine Learning. It focuses on theoretical understanding of diffusion models and accelerating inference of diffusion models. It shares the same potential societal consequences with diffusion models, which is a well established subject in ML. There are many potential societal consequences of our work, none of which we feel must be specifically highlighted here.

## Acknowledgment

This work is supported by funds from NSF-2125511, NSF CCF-2045354 and the Stanford School of Engineering Fellowship.

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

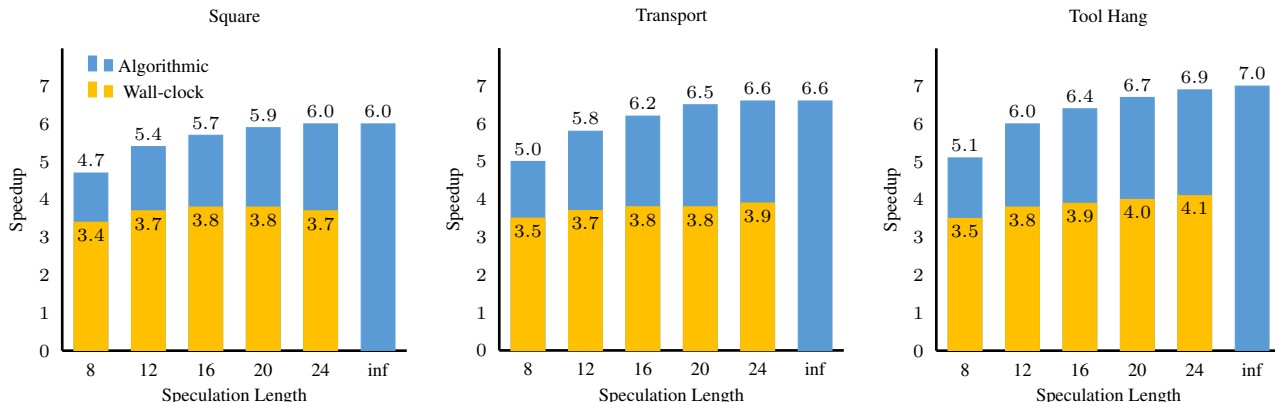

Figure 5: Speedup from ASD of diffusion policies in three Robomimic tasks. Unlike the image generation experiments where we parallelize across multiple GPUs, we only use *one GPU* in these experiments and the parallel verification is achieved through batching.

Eldan, R. Thin shell implies spectral gap up to polylog via a stochastic localization scheme. *Geometric and Functional Analysis*, 23(2):532–569, 2013.

Gupta, S., Cai, L., and Chen, S. Faster diffusion-based sampling with randomized midpoints: Sequential and parallel. *arXiv preprint arXiv:2406.00924*, 2024.

Hessel, J., Holtzman, A., Forbes, M., Bras, R. L., and Choi, Y. Clipscore: A reference-free evaluation metric for image captioning. *arXiv preprint arXiv:2104.08718*, 2021.

Ho, J., Jain, A., and Abbeel, P. Denoising diffusion probabilistic models. *Advances in Neural Information Processing Systems*, 33:6840–6851, 2020.

Kingma, D., Salimans, T., Poole, B., and Ho, J. Variational diffusion models. *Advances in neural information processing systems*, 34:21696–21707, 2021.

Le Gall, J.-F. *Brownian motion, martingales, and stochastic calculus*. Springer, 2016.

Leviathan, Y., Kalman, M., and Matias, Y. Fast inference from transformers via speculative decoding. In *International Conference on Machine Learning*, pp. 19274–19286. PMLR, 2023.

Lou, A., Meng, C., and Ermon, S. Discrete diffusion modeling by estimating the ratios of the data distribution. In *Forty-first International Conference on Machine Learning*.

Lu, C. and Song, Y. Simplifying, stabilizing and scaling continuous-time consistency models. *arXiv preprint arXiv:2410.11081*, 2024.

Lu, C., Zhou, Y., Bao, F., Chen, J., Li, C., and Zhu, J. Dpm-solver++: Fast solver for guided sampling of diffusion probabilistic models. *arXiv preprint arXiv:2211.01095*, 2022.

Mandlekar, A., Xu, D., Wong, J., Nasiriany, S., Wang, C., Kulkarni, R., Fei-Fei, L., Savarese, S., Zhu, Y., and Martín-Martín, R. What matters in learning from offline human demonstrations for robot manipulation. In *arXiv preprint arXiv:2108.03298*, 2021.

Meng, C., Gao, R., Kingma, D. P., Ermon, S., Ho, J., and Salimans, T. On distillation of guided diffusion models. *arXiv preprint arXiv:2210.03142*, 2022.

Montanari, A. Sampling, diffusions, and stochastic localization. *arXiv preprint arXiv:2305.10690*, 2023.

Nichol, A., Dhariwal, P., Ramesh, A., Shyam, P., Mishkin, P., McGrew, B., Sutskever, I., and Chen, M. Glide: Towards photorealistic image generation and editing with text-guided diffusion models. *arXiv preprint arXiv:2112.10741*, 2021.

Oksendal, B. *Stochastic differential equations: an introduction with applications*. Springer Science & Business Media, 2013.

Pokle, A., Geng, Z., and Kolter, J. Z. Deep equilibrium approaches to diffusion models. *Advances in Neural Information Processing Systems*, 35:37975–37990, 2022.

Reuss, M., Li, M., Jia, X., and Lioutikov, R. Goal conditioned imitation learning using score-based diffusion policies. In *Robotics: Science and Systems*, 2023.

Rombach, R., Blattmann, A., Lorenz, D., Esser, P., and Ommer, B. High-resolution image synthesis with latent diffusion models. In *Proceedings of the IEEE/CVF Conference on Computer Vision and Pattern Recognition*, pp. 10684–10695, 2022.

Schuhmann, C., Beaumont, R., Vencu, R., Gordon, C. W., Wightman, R., Cherti, M., Coombes, T., Katta, A., Mullis, C., Wortsman, M., Schramowski, P., Kundurthy, S. R., Crowson, K., Schmidt, L., Kaczmarczyk, R., and Jitsev, J. LAION-5b: An open large-scale dataset for training next generation image-text models. In *Thirty-sixth Conference on Neural Information Processing Systems Datasets and Benchmarks Track*, 2022. URL `https://openreview.net/forum?id=M3Y74vmsMcY`.

Shih, A., Sadigh, D., and Ermon, S. Training and inference on any-order autoregressive models the right way. *Advances in Neural Information Processing Systems*, 35: 2762–2775, 2022.

Shih, A., Belkhale, S., Ermon, S., Sadigh, D., and Anari, N. Parallel sampling of diffusion models. *Advances in Neural Information Processing Systems*, 36, 2024.

Sohl-Dickstein, J., Weiss, E., Maheswaranathan, N., and Ganguli, S. Deep unsupervised learning using nonequilibrium thermodynamics. In *International Conference on Machine Learning*, pp. 2256–2265. PMLR, 2015.

Song, J., Meng, C., and Ermon, S. Denoising diffusion implicit models. In *9th International Conference on Learning Representations, ICLR 2021, Virtual Event, Austria, May 3-7, 2021*, 2021a.

Song, Y., Sohl-Dickstein, J., Kingma, D. P., Kumar, A., Ermon, S., and Poole, B. Score-based generative modeling through stochastic differential equations. In *9th International Conference on Learning Representations, ICLR 2021*, 2021b.

Song, Y., Dhariwal, P., Chen, M., and Sutskever, I. Consistency models. *arXiv preprint arXiv:2303.01469*, 2023.

Van Handel, R. Probability in high dimension. *Lecture Notes (Princeton University)*, 2(3):2–3, 2014.

von Platen, P., Patil, S., Lozhkov, A., Cuenca, P., Lambert, N., Rasul, K., Davaadorj, M., and Wolf, T. Diffusers: State-of-the-art diffusion models. `https://github.com/huggingface/diffusers`, 2022.

Watson, D., Chan, W., Ho, J., and Norouzi, M. Learning fast samplers for diffusion models by differentiating through sample quality. In *International Conference on Learning Representations*, 2022.

## A. Mathematical Preliminaries

In this section, we state some standard results from probability that we use

**Lemma 6** (Pinsker's Inequality (Van Handel, 2014)). *For any two distributions $P$ and $Q$,*

$$\mathsf{TV}\left(P,Q\right) \leq \sqrt{\frac{\mathsf{KL}\left(P\|Q\right)}{2}}$$

**Lemma 7** (Doob's Maximal Inequality (Le Gall, 2016)). *Let $\mathbf{x}_1, \ldots, \mathbf{x}_m$ be an $\mathbb{R}^d$ valued submartingale. Then for any $p > 1$,*

$$\mathbb{E}[\sup_{i \in [m]} \|\mathbf{y}_i - \mathbf{y}_0\|^p] \leq \left(\frac{p}{p-1}\right)^p \mathbb{E}\left[\|\mathbf{y}_m - \mathbf{y}_0\|^p\right]$$

**Lemma 8** (Ito's Lemma (Oksendal, 2013)). *Let $f : \mathbb{R} \times \mathbb{R}^d \to \mathbb{R}$ be a differentiable function and let $\mathbf{x}_t$ be a stochastic process governed by the following SDE:*

$$\mathrm{d}\mathbf{x}_t = a_t \, \mathrm{d}t + \sigma_t dB_t$$

*Then $\mathbf{y}_t = f(t, \mathbf{x}_t)$ satisfies the following SDE:*

$$\mathrm{d}\mathbf{y}_t = (\partial_t f(t, \mathbf{x}_t) + \langle \nabla_\mathbf{x} f(t, \mathbf{x}_t), a_t \rangle + \frac{1}{2}\mathsf{Tr}(\nabla_\mathbf{x}^2 f(t, \mathbf{x}_t)\sigma_t \sigma_t^2)) \, \mathrm{d}t + \langle \nabla f(t, \mathbf{x}_t), \sigma_t \, \mathrm{d}B_t \rangle$$

We also make use of the following result, which is a direct corollary of Girsanov's theorem (Oksendal, 2013) :

**Lemma 9** (Girsanov Bound for KL Divergence). *For $t \in [0, T]$, let $\mathbf{x}_t$ and $\mathbf{y}_t$ be two stochastic processes governed by the following SDEs:*

$$\mathrm{d}\mathbf{x}_t = a_t(\mathbf{x}_{\leq t}) \, \mathrm{d}t + \sigma \, \mathrm{d}B_t$$
$$\mathrm{d}\mathbf{y}_t = b_t(\mathbf{y}_{\leq t}) \, \mathrm{d}t + \sigma \, \mathrm{d}B_t$$

*Let $P^{[0,T]}$ and $Q^{[0,T]}$ denote the path measures of $\mathbf{x}$ and $\mathbf{y}$ in the time interval $[0, T]$. Then,*

$$\mathsf{KL}\left(P^{[0,T]}\|Q^{[0,T]}\right) = \mathsf{KL}\left(\mathbf{x}_0\|\mathbf{y}_0\right) + \frac{\sigma^2}{2}\int_0^T \mathbb{E}_{\mathbf{x}_{0:T} \sim P^{[0,T]}}\left[\|a_t(\mathbf{x}_{\leq t}) - b_t(\mathbf{x}_{\leq t})\|^2\right] \, \mathrm{d}t$$

We also use the following time transformation identity for SDEs, which follows directly from Theorem 8.5.7 of (Oksendal, 2013)

**Lemma 10** (Time Transformation for SDEs). *Let $\mathbf{x}_t$ denote the solution of the following SDE:*

$$\mathrm{d}\mathbf{x}_t = b(t, \mathbf{x}_t) \, \mathrm{d}t + \sigma(t, \mathbf{x}_t) \, \mathrm{d}B_t$$

*and let $r : \mathbb{R}_{\geq 0} \to \mathbb{R}_{\geq 0}$ be a continuously differentiable non-decreasing function with $r(0) = 0$. Then $\mathbf{y}_t = \mathbf{x}_{r(t)}$ satisfies the following SDE:*

$$\mathrm{d}\mathbf{y}_t = b(r(t), \mathbf{y}_t)r'(t) \, \mathrm{d}t + \sigma(r(t), \mathbf{y}_t)\sqrt{r'(t)} \, \mathrm{d}B_t$$

## B. Properties of Stochastic Localization

In this section, we state some additional properties of the Stochastic Localization (SL) process that we use in our analysis, and also present a proof of Theorem 2 in Appendix B.2. Let $\mu$ be a target measure on $\mathbb{R}^d$ whose covariance satisfies $\mathsf{Tr}(\mathsf{Cov}[\mu]) \leq \beta d$ for some $\beta \geq 0$. Recall the SL process from Section 3 defined as follows:

$$\bar{\mathbf{y}}_t = \mathbf{m}(t, \bar{\mathbf{y}}_t) \, \mathrm{d}t + \mathrm{d}B_t, \ \bar{\mathbf{y}}_0 = 0$$
$$\mathbf{m}(t, \mathbf{y}) = \mathbb{E}_{\mathbf{x}^\star \sim \mu, \ \xi \sim \mathcal{N}(0, \mathbf{I})}\left[\mathbf{x}^\star \mid t\mathbf{x}^* + \sqrt{t}\xi = \mathbf{y}\right] \tag{8}$$

In addition, we define $\mu_t$, $\mathbf{m}_t$ and $\Sigma_t$ as follows:

$$\mu_t = \mathsf{Law}_{\mathbf{x}^\star \sim \mu}(\mathbf{x}^\star \,|\mathbf{y}_t) \tag{9}$$

$$\mathbf{m}_t = \mathsf{mean}(\mu_t) \tag{10}$$

$$\Sigma_t = \mathsf{Cov}[\mu_t] \tag{11}$$

It is easy to see that $\mu_0 = \mu$. Our proof of the hidden exchangeability property makes use of the following result, which is proved in El Alaoui & Montanari (2022); Montanari (2023)

**Theorem 11** (Alternate Representation of SL). *The stochastic localization process admits the following alternate representation:*

$$\mathrm{d}\mathbf{y}_t = \mathbf{x}^\star \,\mathrm{d}t + \mathrm{d}W_t$$

*where $\mathbf{x}^\star \sim \mu$ and $W_t$ is a standard Brownian motion on $\mathbb{R}^d$.*

Our analysis also uses the following result on the evolution of $\mathbf{m}_t$ and $\Sigma_t$ which is proved in (Eldan, 2013; Chen, 2021; Chen & Eldan, 2022)

**Theorem 12** (Evolution of Mean and Covariance Along SL). *$\mathbf{m}_t$ and $\Sigma_t$ satisfies the following*

$$\mathrm{d}\mathbf{m}_t = \Sigma_t \,\mathrm{d}W_t$$

$$\frac{\mathrm{d}\,\mathbb{E}[\Sigma_t]}{\mathrm{d}t} = -\,\mathbb{E}[\Sigma_t^2]$$

*where $W_t$ is a Brownian motion on $R^d$ and the expectation is wrt the randomness of the localization process*

To illustrate an application of Theorem 12, we show how to control the error incurred due to Euler discretization of the SL process. An adaptation of this argument in (Benton et al., 2023) was used to analyze the convergence of DDPMs.

**Theorem 13** (Euler Discretization of SL). *Consider any sequence of times $0 \le t_1 \le \cdots \le t_K < \infty$ satisfying $\eta_i = t_{i+1} - t_i \le \eta$. Let $\bar{\mathbf{y}}_t = \mathbf{m}(t, \bar{\mathbf{y}}_t)\,\mathrm{d}t + \mathrm{d}B_t$ denote the SL process and let $P^{\bar{\mathbf{y}}}$ denote its path measure in $[t_1, t_K]$. Define the Euler discretization of the SL process as:*

$$\mathrm{d}\mathbf{y}_t = \mathbf{m}(t_i, \mathbf{y}_{t_i})\,\mathrm{d}t + \mathrm{d}B_t \ , t \in [t_i, t_{i+1}], i \in [K-1]$$

*and let $P^{\mathbf{y}}$ denote its path measure in $[t_1, t_K]$. Suppose the initial distribution $\mu$ has finite second moments and satisfies $\mathsf{Cov}[\mu] \preceq \beta\mathbf{I}$. Then, the KL divergence between the path measures is bounded as $\mathsf{KL}\left(P^{\mathbf{y}}\|P^{\bar{\mathbf{y}}}\right) \le \frac{\eta\beta d}{2}$.*

*Proof.* By Lemma 9,

$$\mathsf{KL}\left(P^{\bar{\mathbf{y}}}\|P^{\mathbf{y}}\right) = \frac{1}{2}\int_{t_0}^{t_K} \mathbb{E}_{\bar{\mathbf{y}}\sim P^{\bar{\mathbf{y}}}}[\|\mathbf{m}_t(\bar{\mathbf{y}}) - \hat{\mathbf{m}}_t(\bar{\mathbf{y}})\|]\,\mathrm{d}t$$

$$= \frac{1}{2}\sum_{i=1}^{K-1}\int_{t_i}^{t_{i+1}} \mathbb{E}_{\bar{\mathbf{y}}\sim P^{\bar{\mathbf{y}}}}[\|\mathbf{m}_t(\mathbf{y}_t) - \mathbf{m}_{t_i}(\mathbf{y}_{t_i})\|]\,\mathrm{d}t$$

$$= \frac{1}{2}\sum_{i=1}^{K-1}\int_{t_i}^{t_{i+1}} \mathbb{E}_{\bar{\mathbf{y}}\sim P^{\bar{\mathbf{y}}}}\left[\left\|\int_{t_i}^{t}\Sigma_s\,\mathrm{d}W_s\right\|^2\right]\,\mathrm{d}t$$

$$= \frac{1}{2}\sum_{i=1}^{K-1}\int_{t_i}^{t_{i+1}}\int_{t_i}^{t} \mathbb{E}[\mathsf{Tr}(\Sigma_s^2)]\,\mathrm{d}s\,\mathrm{d}t$$

where the third step uses Theorem 12 and the last step applies the Ito isometry. From Theorem 12, we note that $\frac{d\mathsf{Tr}(\mathbb{E}[\Sigma_t])}{\mathrm{d}t} =$

$-\mathbb{E}[\mathsf{Tr}(\Sigma_t)^2] \preceq 0$, which implies the following:

$$
\begin{aligned}
\mathsf{KL}\left(P^{\bar{\mathbf{y}}}\|P^{\mathbf{y}}\right) &= \frac{1}{2}\sum_{i=1}^{K-1}\int_{t_i}^{t_{i+1}}\int_{t_i}^{t}\mathbb{E}[\mathsf{Tr}(\Sigma_s^2)]\,\mathrm{d}s\,\mathrm{d}t \\
&= \frac{1}{2}\sum_{i=1}^{K-1}\int_{t_i}^{t_{i+1}}\mathsf{Tr}(\mathbb{E}[\Sigma_{t_i}]-\mathbb{E}[\Sigma_t])\,\mathrm{d}t \\
&\leq \frac{1}{2}\sum_{i=1}^{K-1}(t_{i+1}-t_i)\mathsf{Tr}(\mathbb{E}[\Sigma_{t_i}-\Sigma_{t_{i+1}}]) \\
&\leq \frac{hd}{2}\mathbb{E}[\mathsf{Tr}(\Sigma_{t_1})] \leq \frac{\eta\beta d}{2}
\end{aligned}
$$

where we use the fact that $\mathbb{E}[\Sigma_t]$ is a non-increasing function (in the PSD order). $\qquad\square$

Our adaptive complexity analysis of Algorithm 1 relies on an argument similar to the proof of the above theorem.

## B.1. Equivalence between DDPM and SL

The following result by (Montanari, 2023) proves that the OU Process defined in Section 3 is equivalent to Stochastic Localization

**Theorem 14** (Equivalence of OU Process and SL). *Let $\bar{\mathbf{y}}_t$ denote the SL process as defined above and let $\mathbf{z}_t$ denote the OU process defined by the following SDE:*

$$
d\mathbf{z}_t = -\mathbf{z}_t dt + \sqrt{2}\,\mathrm{d}B_t
$$

*Then, the following holds:*

$$
\bar{\mathbf{y}}_t \overset{d}{=} te^{s(t)}\mathbf{z}_{s(t)}
$$

*where $s(t) = \frac{1}{2}\ln(1+{}^1\!/t)$*

Equipped with the above result, we now present a proof of Theorem 1 below

### B.1.1. PROOF OF THEOREM 1

*Proof.* Let $\mathbf{x}_t^{\leftarrow}$ denote the DDPM forward process as defined in equation (1), which satisfies the following SDE :

$$
\mathrm{d}\mathbf{x}_t^{\rightarrow} = h(t)\mathbf{x}_t^{\rightarrow}\,\mathrm{d}t + \sqrt{u(t)}dW_t, \mathbf{x}_0^{\rightarrow} \sim \mu
$$

Now, let $\mathbf{z}_t$ denote the following OU process:

$$
\mathbf{z}_t = -\mathbf{z}_t\,\mathrm{d}t + \sqrt{2}\,\mathrm{d}B_t, \ \mathbf{z}_0 = \mathbf{x}_0^{\rightarrow} \tag{12}
$$

We now define the functions $\alpha(t)$ and $r(t)$ as follows:

$$
\alpha(t) = \frac{1}{2}\ln\left(1+\int_0^t u(\tau)\exp(-2\int_0^\tau h(s)\,\mathrm{d}s)\,\mathrm{d}\tau\right)
$$

$$
r(t) = \exp(\alpha(t)+\int_0^t h(\tau)\,\mathrm{d}\tau)
$$

Since $u(t) \geq 0$, it is easy to check that $\alpha(t)$ is a non-decreasing function with $\alpha(0) = 0$. Furthermore, $r(t) > 0$ with $r(0) = 1$. We shall now prove that $\mathbf{x}_t^{\rightarrow} = r(t)\mathbf{z}_{\alpha(t)}$. Note that our claim holds for $t = 0$ since $\mathbf{x}_0^{\rightarrow} = \mathbf{z}_0 = r(0)\mathbf{z}_{\alpha(0)}$.

To prove this claim for $t > 0$, we show that $\mathbf{z}_t^{\rightarrow}$ and $r(t)\mathbf{z}_{\alpha(t)}$ follow the same SDE. Since $\alpha(t)$ satisfies the conditions of Lemma 10, we conclude the following:

$$
\mathrm{d}(\mathbf{z}_{\alpha(t)}) = -\alpha'(t)\mathbf{z}_{\alpha(t)}\,\mathrm{d}t + \sqrt{2\alpha'(t)}\,\mathrm{d}B_t
$$

Using the above and Ito's Lemma, we obtain the following:

$$\mathrm{d}(r(t)\mathbf{z}_{\alpha(t)}) = \left[ r'(t) - \alpha'(t)r(t) \right] \mathbf{z}_{\alpha(t)} \, \mathrm{d}t + r(t)\sqrt{2\alpha'(t)} \, \mathrm{d}B_t$$
$$= \left[ \frac{r'(t)}{r(t)} - \alpha'(t) \right] r(t)\mathbf{z}_{\alpha(t)} \, \mathrm{d}t + \sqrt{2\alpha'(t)r(t)^2} \, \mathrm{d}B_t$$

Taking derivatives of $\ln r(t)$ and $e^{2\alpha(t)}$, it is easy to see that the $\frac{r'(t)}{r(t)} = \alpha'(t) + h(t)$ and $2\alpha'(t)r(t)^2 = u(t)$, i.e.,

$$\mathrm{d}(r(t)\mathbf{z}_{\alpha(t)}) = h(t)r(t)\mathbf{z}_{\alpha(t)} \, \mathrm{d}t + \sqrt{u(t)} \, \mathrm{d}B_t$$

From Eq. (12) and the above, we note that $\mathbf{x}_t^{\rightarrow}$ and $r(t)\mathbf{z}_{\alpha(t)}$ satisfy the same SDE. Since $\mathbf{x}_0^{\rightarrow} = r(0)\mathbf{z}_{\alpha(0)}$, we conclude that $\mathbf{x}_t^{\rightarrow} = r(t)\mathbf{z}_{\alpha(t)}$. Then, by definition of the reverse process,

$$\mathbf{z}_t = \frac{\mathbf{x}_{\alpha^{-1}(t)}^{\rightarrow}}{r(\alpha^{-1}(t))} = \frac{\mathbf{x}_{T-\alpha^{-1}(t)}^{\leftarrow}}{r(\alpha^{-1}(t))}$$

From Theorem 14, we know that $\mathbf{y}_t \overset{d}{=} te^{s(t)}\mathbf{z}_{s(t)}$ where where $s(t) = \frac{1}{2}\ln(1 + 1/t)$. Substituting this into the above identity gives us the following:

$$\bar{\mathbf{y}}_t \overset{d}{=} \gamma(t)\mathbf{x}_{\zeta(t)}^{\leftarrow}$$
$$\gamma(t) = \frac{te^{s(t)}}{r(\alpha^{-1}(s(t)))}$$
$$\zeta(t) = T - \alpha^{-1}(s(t))$$

$\square$

## B.2. Proof of Theorem 2

*Proof.* By Theorem 11, we know that the stochastic localization process satisfies:

$$\bar{\mathbf{y}}_t = t\mathbf{x}^* + W_t$$

where $\mathbf{x}^* \sim \mu$ and $W_t$ is a standard Brownian motion on $\mathbb{R}^d$. By the properties of Brownian increments, the following holds for any $i \in [m-1]$ and $\pi \in \mathbb{S}_{m-1}$

$$\mathsf{Law}((\bar{\mathbf{y}}_{t_i+\eta_i} - \bar{\mathbf{y}}_{t_i})_{i\in[m-1]}|\mathbf{x}^*) = \bigotimes_{i\in[m-1]} \mathcal{N}(\eta_i\mathbf{x}^*, \eta_i\mathbf{I})$$
$$= \mathsf{Law}((\mathbf{y}_{t_{\pi(i)}+\eta_i} - \mathbf{y}_{t_{\pi(i)}})_{i\in[m-1]}|\mathbf{x}^*)$$

By marginalizing $\mathbf{x}^*$, we complete the proof of the first claim. The second claim directly follows by setting the time increments $\eta_i$ to be equal. $\square$

# C. Theoretical Guarantees for Auto-Speculative Decoding

In this section, we present our theoretical guarantees for Auto-Speculative decoding. To begin with, we prove the correctness of the Gaussian Rejection Sampler in Algorithm 3. We note that this result implies that the Verifier in Algorithm 2 always outputs samples $(\mathbf{z}_{a+1}, \ldots, \mathbf{z}_b) \sim q(.|\mathbf{y}_{\leq a})$ that are distributed according to the conditional target distribution.

## C.1. Correctness of Algorithm 3

**Theorem 15.** *Let $\xi \sim \mathcal{N}(0, \mathbf{I})$. Then, for any $\hat{\mathbf{m}}, \mathbf{m} \in \mathbb{R}^d$ and $\sigma > 0$, GRS$(u, \xi, \hat{\mathbf{m}}, \mathbf{m})$ outputs $(\mathbf{x}, b)$ such that $\mathbf{x} \sim \mathcal{N}(\mathbf{m}, \sigma^2\mathbf{I})$ and $\mathbb{P}[b = \mathsf{False}] = \mathsf{TV}\left(\mathcal{N}(\hat{\mathbf{m}}, \sigma^2\mathbf{I}), \mathcal{N}(\mathbf{m}, \sigma^2\mathbf{I})\right)$*

*Proof.* Without loss of generality, we consider $\sigma = 1$ and observe that

$$\mathbb{P}[b = \mathsf{True}] = \int \min\left(1, \frac{\mathcal{N}(\xi + \mathbf{v}|0, \mathbf{I})}{\mathcal{N}(\xi|0, \mathbf{I})}\right) \mathcal{N}(\xi|0, \mathbf{I}) \, d\xi$$

$$= \int \min(\mathcal{N}(\xi|0, \mathbf{I}), \mathcal{N}(\xi + \mathbf{v}|0, \mathbf{I})) \, d\xi$$

$$= \int \min(\mathcal{N}(\xi|\mathbf{m}, \mathbf{I}), \mathcal{N}(\xi|\hat{\mathbf{m}}, \mathbf{I})) \, d\xi$$

$$= 1 - \mathsf{TV}\left(\mathcal{N}(\xi|\mathbf{m}, \mathbf{I}), \mathcal{N}(\xi|\hat{\mathbf{m}}, \mathbf{I})\right)$$

which proves the second claim. Now, let $\mathbf{g} = \mathbf{x} - \mathbf{m}$ and $\mathbf{e}_1, \ldots, \mathbf{e}_d$ be an orthonormal basis of $\mathbb{R}^d$ with $\mathbf{v} = \|\mathbf{v}\|\mathbf{e}_1$. Then,

$$\mathbf{g} = \begin{cases} \xi + \mathbf{v} & \text{w.p. } \min\left(1, \frac{\mathcal{N}(\xi+\mathbf{v}|0,\mathbf{I})}{\mathcal{N}(\xi|0,\mathbf{I})}\right) \\ (\mathbf{I} - 2\mathbf{e}_1\mathbf{e}_1^T)\mathbf{v} & \text{w.p. } \max\left(0, 1 - \frac{\mathcal{N}(\xi+\mathbf{v}|0,\mathbf{I})}{\mathcal{N}(\xi|0,\mathbf{I})}\right) \end{cases} \tag{13}$$

Then, the law of $\mathbf{g}$ satisfies the following:

$$\mathsf{Law}(\mathbf{g}) = \int \delta_{\xi+\mathbf{v}}(\mathbf{g}) \min\left(\mathcal{N}(\xi|0, \mathbf{I}), \mathcal{N}(\xi + \mathbf{v}|0, \mathbf{I})\right) d\xi$$

$$+ \int \delta_{(\mathbf{I}-2\mathbf{e}\mathbf{e}^T)\xi}(\mathbf{g}) \max\left(\mathcal{N}(\xi|0, \mathbf{I}), \mathcal{N}(\xi|0, \mathbf{I}) - \mathcal{N}(\xi + \mathbf{v}|0, \mathbf{I})\right) d\xi$$

$$= \min\left(\mathcal{N}(\mathbf{g} - \mathbf{v}|0, \mathbf{I}), \mathcal{N}(\mathbf{g}|0, \mathbf{I})\right) + \int \delta_{(\mathbf{I}-2\mathbf{e}\mathbf{e}^T)\xi}(\mathbf{g}) \max\left(0, \mathcal{N}(\xi|0, \mathbf{I}) - \mathcal{N}(\xi + \mathbf{v}|0, \mathbf{I})\right) d\xi \tag{14}$$

where $\delta_{\mathbf{x}}$ denotes the Dirac measure supported at $\mathbf{x}$. To bound the integral on the RHS, we note that the matrix $\mathbf{M} = \mathbf{I} - 2\mathbf{e}_1\mathbf{e}_1^T$ is a reflection operator along the $\mathbf{e}_1$ axis, i.e., for any $\mathbf{x} \in \mathbb{R}^d$ and $\mathbf{y} = \mathbf{M}\mathbf{x}$, $\langle \mathbf{y}, \mathbf{e}_1 \rangle = -\langle \mathbf{x}, \mathbf{e}_1 \rangle$ and $\langle \mathbf{y}, \mathbf{e}_j \rangle - \langle \mathbf{x}, \mathbf{e}_j \rangle$ for any $j \neq 1$. Hence, it follows that $\|\mathbf{y}\| = \|\mathbf{x}\|$ and $\mathbf{x} = \mathbf{M}\mathbf{y}$

$$\int \delta_{\mathbf{M}\xi}(\mathbf{g}) \max\left(0, \mathcal{N}(\xi|0, \mathbf{I}) - \mathcal{N}(\xi + \mathbf{v}|0, \mathbf{I})\right) d\xi = \max\left(0, \mathcal{N}(\mathbf{M}\mathbf{g}|0, \mathbf{I}) - \mathcal{N}(\mathbf{M}\mathbf{g} + \mathbf{v}|0, \mathbf{I})\right) \tag{15}$$

Now, $\mathcal{N}(\mathbf{M}\mathbf{g}|0, \mathbf{I}) = \mathcal{N}(\mathbf{g}|0, \mathbf{I})$ since $\|\mathbf{M}\mathbf{g}\| = \|\mathbf{g}\|$. Moreover, since $\mathbf{M} = \mathbf{M}^T$

$$\|\mathbf{M}\mathbf{g} + \mathbf{v}\|^2 = \|\mathbf{M}\mathbf{g}\|^2 + \|\mathbf{v}\|^2 + 2\langle \mathbf{M}\mathbf{g}, \mathbf{v} \rangle$$

$$= \|\mathbf{g}\|^2 + \|\mathbf{v}\|^2 + 2\langle \mathbf{g}, \mathbf{M}\mathbf{e}_1 \rangle \|\mathbf{v}\|$$

$$= \|\mathbf{g}\|^2 + \|\mathbf{v}\|^2 - 2\langle \mathbf{g}, \mathbf{e}_1 \rangle \|\mathbf{v}\|$$

$$= \|\mathbf{g} - \mathbf{v}\|$$

where we use the fact that $\mathbf{M}\mathbf{e}_1 = -\mathbf{e}_1$ and $\mathbf{v} = \|\mathbf{v}\|\mathbf{e}_1$. Hence, $\mathcal{N}(\mathbf{M}\mathbf{g} + \mathbf{v}|0, \mathbf{I}) = \mathcal{N}(\mathbf{g} - \mathbf{v}|0, \mathbf{I})$. Substituting into equation (15), we have:

$$\int \delta_{\mathbf{M}\xi}(\mathbf{g}) \max\left(0, \mathcal{N}(\xi|0, \mathbf{I}) - \mathcal{N}(\xi + \mathbf{v}|0, \mathbf{I})\right) d\xi = \max\left(0, \mathcal{N}(\mathbf{g}|0, \mathbf{I}) - \mathcal{N}(\mathbf{g} - \mathbf{v}|0, \mathbf{I})\right)$$

Substituting this into (14), we get

$$\mathsf{Law}(\mathbf{g}) = \min\left(\mathcal{N}(\mathbf{g} - \mathbf{v}|0, \mathbf{I}), \mathcal{N}(\mathbf{g}|0, \mathbf{I})\right) + \max\left(0, \mathcal{N}(\mathbf{g}|0, \mathbf{I}) - \mathcal{N}(\mathbf{g} - \mathbf{v}|0, \mathbf{I})\right)$$
$$= \mathcal{N}(\mathbf{g}|0, \mathbf{I})$$

Since $\mathbf{x} = \mathbf{m} + \mathbf{g}$, $\mathbf{x} \sim \mathcal{N}(\mathbf{m}, \mathbf{I})$ which completes the proof. □

## C.2. Proof of Theorem 4

In this section, we present our analysis of Autospeculation for the Stochastic Localization Process. We use $a_t$ to denote the value of the index $a$ at the *end* of the $t^{\text{th}}$ iteration in Algorithm 1.

**Lemma 16.** $(\mathbf{y}_0, \ldots, \mathbf{y}_{a_t})$ *is distributed correctly according to the target distribution* $q(\mathbf{y}_0, \ldots, \mathbf{y}_{a_t})$. *Furthermore,* $a_t$ *is a strictly increasing sequence in* $t$

*Proof.* We prove this claim by induction. Clearly, the claim holds for $t = 0$. Now, suppose it holds for some $t$. By definition of the proposal means and the target means in equations (7) and (6), $\hat{\mathbf{m}}_{a_t+1} = \mathbf{m}_{a_t+1}$. Then, by Theorem 15, Algorithm 2 does not reject the index $a_t + 1$, thereby ensuring that $a_{t+1} > a_t + 1$ as per lines 13-18 of Algorithm 1. Since the verifier's outputs are always distributed according to the conditional target distribution, $\mathsf{Law}(\mathbf{y}_{a_t+1}, \ldots, \mathbf{y}_{a_{t+1}} | \mathbf{y}_0, \ldots, \mathbf{y}_{a_t}) = q(\mathbf{y}_{a_t+1} | \mathbf{y}_0, \ldots, \mathbf{y}_{a_t})$. The proof is completed by removing the conditioning on $\mathbf{y}_0, \ldots, \mathbf{y}_{a_t}$ by applying the induction hypothesis.

$\square$

Since Algorithm 1 terminates when $a \geq K$, Lemma 16 implies Theorem 4

**C.3. Adaptive Complexity of Algorithm 1**

In this section, we analyze the adaptive complexity of Algorithm 1 for the SL process. Our proof combines arguments from Anari et al. (2024a) with a careful analysis of speculations of the SL process motivated by the hidden exchangeability property. We first define the $a_i^\star(u_{1:K}, \xi_{1:K})$ as the maximum possible value of $a$ such that the parallel rejection sampler does not accept the $i^{\text{th}}$ proposal. Formally

$$a_i^\star(u_{1:K}, \xi_{1:K}) = \max \left\{ a \in [K] \mid \mathsf{Verifier}(u_{a+1:K}, \xi_{a+1:K}, \hat{\mathbf{m}}_{a+1:K}, \mathbf{m}_{a+1:K}) < i \right\}$$

where $\mathbf{m}_i, \hat{\mathbf{m}}_i$ are defined as in Algorithm 1 (with $b = K$). We use $R$ to denote the round complexity (i.e. the number of iterations) taken by Algorithm 1. Since Algorithm 1 makes exactly two parallel model calls per iteration, bounding $R$ is equivalent to bounding the adaptive complexity. We first prove a worst case bound on $R$ as a function of $a_i^\star$

**Lemma 17.** *The following holds for any integer* $\theta \in \mathbb{N}$, $u_{1:K} \in [0,1]^K$ *and* $\sigma_{1:K} \in \mathbb{R}^K$.

$$R \leq 1 + \frac{K}{\theta} + |\{i \in [K] \mid a_i^\star(u_{1:K}, \xi_{1:K}) \geq i - \theta\}|$$

*Proof.* Let $t$ be denoted a good iteration if $a_t - a_{t-1} > \theta$ and a bad iteration otherwise. Since the algorithm halts whenever $a \geq K$, the number of good iterations is bounded by $\frac{K}{\theta} + 1$. Since $a_t$ is a monotonically increasing sequence by Lemma 16, any bad iteration satisfies $a_{a_t}^\star(u_{1:K}, \xi_{1:K}) \geq a_{t-1} \geq a_t - \theta$, i.e., $a_t \in \{i \in [K] \mid a_i^\star(u_{1:K}, \xi_{1:K}) \geq i - \theta\}$. Since $t \to a_t$ is a bijection, we conclude that the number of bad rounds is upper bounded by $|\{i \in [K] \mid a_i^\star(u_{1:K}, \xi_{1:K}) \geq i - \theta\}|$ $\square$

We now use the above Lemma and the time-invariance properties of the SL process to prove an expectation bound on the round complexity, which establishes the first claim of Theorem 5

**Theorem 18** (Expected Round Complexity of Algorithm 1). *Under the assumptions and parameter settings of Theorem 5, Algorithm 1 run for the discretization of the SL process with* $\theta \asymp (K/\beta\eta d)^{1/3}$ *makes* $O(K^{2/3}(\beta d\eta)^{1/3})$ *parallel model calls in expectation.*

*Proof.* From Lemma 17, we note that

$$\mathbb{E}[R] \leq 1 + \frac{K}{\theta} + \sum_{i=\theta}^{K} \mathbb{P}[a_i^\star(u_{1:K}, \xi_{1:K}) \geq i - \theta] \tag{16}$$

Note that $a_i^\star(u_{1:K}, \xi_{1:K}) \geq i - \theta$ implies that there exists some $a \in [i - \theta, i - 1]$ such that the proposal for step $i$ made at step $a$ was rejected by the Verifier. Then, by Theorem 15,

$$\sum_{i=\theta}^{K} \mathbb{P}[a_i^\star(u_{1:K}, \xi_{1:K}) \geq i - \theta] \leq \sum_{i \geq \theta} \mathbb{E}_{u,\xi} \left[ \max_{a \in [i-\theta, i-1]} \mathsf{TV}\left(q(\mathbf{y}_i | \mathbf{y}_a), p(\mathbf{y}_i | \mathbf{y}_a)\right) \right]$$

$$\leq \sqrt{K} \sqrt{\sum_{i \geq \theta} \mathbb{E}_{u,\xi} \left[ \max_{a \in [i-\theta, i-1]} \mathsf{TV}\left(q(\mathbf{y}_i | \mathbf{y}_a), p(\mathbf{y}_i | \mathbf{y}_a)\right)^2 \right]} \tag{17}$$

By our choice of the proposal and target distributions, $p(\mathbf{y}_i|\mathbf{y}_a)$ and $q(\mathbf{y}_i|\mathbf{y}_a)$ are Gaussians with the same variance. To this end, their TV distance is an increasing concave function of the difference of their means. Also, note that for $a \in [i - \theta, i - 1]$ $p(\mathbf{y}_i|\mathbf{y}_a)$ and $q(\mathbf{y}_i|\mathbf{y}_a)$ are Doob Martingales with respect to the filtration generated by $(u_j, \xi_j)_{j \leq i}$, we conclude via the Doob Maximal Inequality for $p = 2$

$$\mathbb{E}_{u,\xi}\left[\max_{a \in [i-\theta, i-1]} \mathsf{TV}\left(q(\mathbf{y}_i|\mathbf{y}_a), p(\mathbf{y}_i|\mathbf{y}_a)\right)^2\right] \lesssim \mathbb{E}\left[\mathsf{TV}\left(q(\mathbf{y}_i|\mathbf{y}_{i-\theta}), p(\mathbf{y}_i|\mathbf{y}_{i-\theta})\right)^2\right] \tag{18}$$

We now bound the above quantity for each index $i$. To this end, let $l = i - \theta$ and define the following stochastic processes associated with the proposal and target.

$$d\mathbf{y}_{t_l+t}^{(\mathsf{P})} = \mathbf{m}(t_l, \mathbf{y}_{t_l}^{(\mathsf{P})})\, dt + dB_{t_l+t}$$
$$d\mathbf{y}_{t_i+t}^{(\mathsf{T})} = \mathbf{m}(t_i, \mathbf{y}_{t_i}^{(\mathsf{T})})\, dt + dB_{t_i+t}$$

Then, by the data processing inequality and equation (18), we obtain:

$$\mathbb{E}\left[\max_{a \in [i-\theta, i-1]} \mathsf{TV}\left(q(\mathbf{y}_i|\mathbf{y}_a), p(\mathbf{y}_i|\mathbf{y}_a)\right)^2\right] \lesssim \mathbb{E}\left[\mathsf{TV}\left(P^{\mathbf{y}^{(\mathsf{P})}}, P^{\mathbf{y}^{(\mathsf{T})}}\right)^2\right] \tag{19}$$

We now define the following auxiliary processes:

$$d\bar{\mathbf{y}}_{t_l+t}^{(\mathsf{P})} = \mathbf{m}(t_l + t, \bar{\mathbf{y}}_{t_l+t}^{(\mathsf{P})})\, dt + dB_{t_l+t}$$
$$d\bar{\mathbf{y}}_{t_i+t}^{(\mathsf{T})} = \mathbf{m}(t_i + t, \bar{\mathbf{y}}_{t_l+t}^{(\mathsf{P})})\, dt + dB_{t_i+t}$$

Similarly, let $P^{\bar{\mathbf{y}}^{(\mathsf{P})}}$ and $P^{\bar{\mathbf{y}}^{(\mathsf{T})}}$ denote their respective path measures for $t \in [0, \eta_l]$. We observe that: 1. The differential increments of $\bar{\mathbf{y}}_t^{(\mathsf{P})}$ and $\bar{\mathbf{y}}_t^{(\mathsf{T})}$ match that of the SL process modulo a time-shift, 2. $\mathbf{y}_t^{(\mathsf{P})}$ and $\mathbf{y}_t^{(\mathsf{T})}$ are Euler discretizations of $\bar{\mathbf{y}}^{(\mathsf{P})}$ and $\bar{\mathbf{y}}^{(\mathsf{T})}$ respectively. By Pinsker's and Cauchy Schwarz Inequality,

$$\mathbb{E}\left[\mathsf{TV}\left(P^{\mathbf{y}^{(\mathsf{P})}}, P^{\mathbf{y}^{(\mathsf{T})}}\right)^2\right] \lesssim \mathbb{E}\left[\mathsf{TV}\left(P^{\mathbf{y}^{(\mathsf{P})}}, P^{\bar{\mathbf{y}}^{(\mathsf{P})}}\right)^2 + \mathsf{TV}\left(P^{\mathbf{y}^{(\mathsf{T})}}, P^{\bar{\mathbf{y}}^{(\mathsf{T})}}\right)^2 + \mathsf{TV}\left(P^{\bar{\mathbf{y}}^{(\mathsf{P})}}, P^{\bar{\mathbf{y}}^{(\mathsf{T})}}\right)^2\right]$$
$$\lesssim \mathbb{E}\left[\mathsf{KL}\left(P^{\bar{\mathbf{y}}^{(\mathsf{P})}}\|P^{\mathbf{y}^{(\mathsf{P})}}\right) + \mathsf{KL}\left(P^{\mathbf{y}^{(\mathsf{T})}}\|P^{\bar{\mathbf{y}}^{(\mathsf{T})}}\right) + \mathsf{KL}\left(P^{\bar{\mathbf{y}}^{(\mathsf{P})}}\|P^{\bar{\mathbf{y}}^{(\mathsf{T})}}\right)\right] \tag{20}$$

To upper bound $\mathsf{KL}\left(P^{\bar{\mathbf{y}}^{(\mathsf{P})}}\|P^{\mathbf{y}^{(\mathsf{P})}}\right)$, we use the fact that $\mathbf{y}^{(\mathsf{P})}$ corresponds to the Euler discretization of $\bar{\mathbf{y}}^{(\mathsf{P})}$ and follow the same steps as the proof of Theorem 13:

$$\mathbb{E}[\mathsf{KL}\left(P^{\bar{\mathbf{y}}^{(\mathsf{P})}}\|P^{\mathbf{y}^{(\mathsf{P})}}\right)] \lesssim \int_{t_l}^{t_l+\eta_l} \mathbb{E}[\|\mathbf{m}_t - \mathbf{m}_{t_l}\|^2]\, dt$$
$$\lesssim \int_{t_l}^{t_l+\eta_l} \int_{t_l}^{t} \mathbb{E}[\mathsf{Tr}(\Sigma_s^2)] ds\, dt$$
$$\lesssim \int_{t_l}^{t_l+\eta_l} \mathsf{Tr}(\mathbb{E}[\Sigma_{t_l}] - \mathbb{E}[\Sigma_t])$$
$$\lesssim \eta_{i-\theta}\mathsf{Tr}(\mathbb{E}[\Sigma_{t_{i-\theta}}] - \mathbb{E}[\Sigma_{t_{i-\theta+1}}]) \tag{21}$$

By a similar computation, we also have

$$\mathbb{E}[\mathsf{KL}\left(P^{\bar{\mathbf{y}}^{(\mathsf{T})}}\|P^{\mathbf{y}^{(\mathsf{T})}}\right)] \leq \eta_{i-\theta}\mathsf{Tr}(\mathbb{E}[\Sigma_{t_i}] - \mathbb{E}[\Sigma_{t_i+\eta_{i-\theta}}]) \tag{22}$$

Using the fact that the increments of $\bar{\mathbf{y}}^{(\mathsf{P})}$ and $\bar{\mathbf{y}}^{(\mathsf{T})}$ are time-shifted versions of the SL process increments via Girsanov's

theorem and Theorem 12:

$$\mathbb{E}[\mathsf{KL}\left(P^{\bar{\mathbf{y}}^{(\mathrm{P})}}\|P^{\bar{\mathbf{y}}^{(\mathrm{T})}}\right)] \lesssim \int_0^{\eta_l} \mathbb{E}[\|\mathbf{m}_{t_l+t} - \mathbf{m}_{t_i+t}\|^2] \, \mathrm{d}t$$

$$\lesssim \int_0^{\eta_l} \int_{t_l+t}^{t_i+t} \mathbb{E}[\mathsf{Tr}(\Sigma_s^2)] \, \mathrm{d}s \, \mathrm{d}t$$

$$\lesssim \int_0^{\eta_l} \mathsf{Tr}(\mathbb{E}[\Sigma_{t_l+t}] - \mathbb{E}[\Sigma_{t_i+t}]) \, \mathrm{d}t$$

$$\lesssim \eta_{i-\theta} \mathsf{Tr}(\mathbb{E}[\Sigma_{t_{i-\theta}}] - \mathbb{E}[\Sigma_{t_i+\eta_{i-\theta}}]) \qquad (23)$$

where we use the fact that $\mathbb{E}[\Sigma_t]$ is non-increasing in the PSD order.

Substituting equations (20) (21), (22) and (23) into equation (19) and summing over $i$, we obtain the following:

$$\mathbb{E}\left[\max_{a \in [i-\theta, i-1]} \mathsf{TV}\left(q(\mathbf{y}_i|\mathbf{y}_a), p(\mathbf{y}_i|\mathbf{y}_a)\right)^2\right] \lesssim \sum_{i \geq \theta} \eta_{i-\theta} \mathsf{Tr}(\mathbb{E}[\Sigma_{t_{i-\theta}}] - \mathbb{E}[\Sigma_{t_{i-\theta+1}}]) + \sum_{i \geq \theta} \eta_{i-\theta} \mathsf{Tr}(\mathbb{E}[\Sigma_{t_i}] - \mathbb{E}[\Sigma_{t_i+\eta_{i-\theta}}])$$

$$+ \sum_{i \geq \theta} \eta_{i-\theta} \mathsf{Tr}(\mathbb{E}[\Sigma_{t_{i-\theta}}] - \mathbb{E}[\Sigma_{t_i+\eta_{i-\theta}}])$$

$$\lesssim 2\eta \mathsf{Tr}(\mathbb{E}[\Sigma_0]) + \theta h \mathsf{Tr}(\mathbb{E}[\Sigma_0]) \lesssim \theta \eta \beta d \qquad (24)$$

Substituting (24) in (17) and (16), we conclude the following:

$$\mathbb{E}[R] \lesssim \frac{K}{\theta} + \sum_{i \geq \theta} \mathsf{TV}\left(P^{\mathbf{y}^{(\mathrm{P})}}, P^{\mathbf{y}^{(\mathrm{T})}}\right)$$

$$\lesssim \frac{K}{\theta} + \sqrt{K}\sqrt{\sum_{i \geq \theta} \mathbb{E}\left[\mathsf{TV}\left(P^{\mathbf{y}^{(\mathrm{P})}}, P^{\mathbf{y}^{(\mathrm{T})}}\right)^2\right]}$$

$$\lesssim \frac{K}{\theta} + \sqrt{K \theta \eta \beta d}$$

$\theta = (K/\eta\beta d)^{1/3}$, we obtain the desired expected round complexity. $\qquad\square$

We now boost the expected round complexity guarantee of Theorem Theorem 18 into a high probability guarantee, thereby proving the second claim of Theorem 5. The proof adapts the arguments of (Anari et al., 2024a), Theorem 28.

**Theorem 19** (High Probability Bound for Round Complexity). *Let $\delta \in (0,1)$ be arbitrary. Consider Algorithm 1 for the SL process run under the assumptions and parameter settings of Theorem 18. Then, with probability at least $1 - \delta$, Algorithm 1 makes at most $O(K^{2/3}(\beta d h)^{1/3} \ln(1/\delta))$ parallel model calls.*

*Proof.* By Theorem 18 and Markov's inequality, there exists a constant $M$ such that $\mathbb{P}[R > MK^{2/3}(\eta\beta d)^{1/3}] \leq 1/2$. We shall now prove via induction that the following holds for any integer $c \geq 1$ and $K \geq 1$:

$$\mathbb{P}[R > cMK^{2/3}(\eta\beta d)^{1/3}] \leq 2^{-c}$$

The case $c = 1$ holds via Markov's inequality and the case $K = 1$ is trivially true.

Now, suppose the statement holds for any $K < K_1$ and consider an instance with $K = K_1$. For $i \in [K-1]$ let $E_i$ be the event that Algorithm 1 doesn't terminate and has $a = i$ after running for $MK^{2/3}(\eta\beta d)^{1/3}$ rounds. Clearly, $\sum_{i \leq K-1} \mathbb{P}[E_i] \leq 1/2$. Moreover, by Lemma 16, running the algorithm for $MK^{2/3}(\eta\beta d)^{1/3}$ ensures $a \geq MK^{2/3}(\eta\beta d)^{1/3}$. Hence, $\mathbb{P}[E_0] = 0$ and for any $c > 1$:

$$\mathbb{P}[R > cMK^{2/3}(\eta\beta d)^{1/3}] = \sum_{i \geq K-1} \mathbb{P}[R > cMK^{2/3}(\eta\beta d)^{1/3}|E_i]\,\mathbb{P}[E_i]$$

$$\leq 1/2 \max_{i \in [K-1]} \mathbb{P}[R > cMK^{2/3}(\eta\beta d)^{1/3}|E_i] \qquad (25)$$

Note that for any $i \in [K-1]$, $E_i$ is measurable w.r.t the filtration generated by $(u_j, \xi_j)_{j \leq i}$, i.e., whether or not $E_i$ occurs is determined exactly by these random variables variables, and thus $E_i$ is independent of $(u_l, \xi_l)_{l > i}$. Furthermore, if $E_i$ occurs Algorithm 1 fixes the random variables $(u_j, \xi_j)_{j \leq i}$ and moves forward with $a = i$, Therefore, the law of the remaining iterations is equal to that of a fresh run of the algorithm on the iterates $(\mathbf{y}_l)_{l > i}$ conditioned on the iterates $(\mathbf{y}_j)_{j \leq i}$ being fixed to their current values (as determined by the $(u_j, \xi_j)_{j \leq i}$). By our induction hypothesis for and the definition of $E_i$, the number of remaining rounds (upon conditioning on $E_i$) exceeds $(c-1)MK^{2/3}(\eta\beta d)^{1/3}$ with probability at most $2^{-c+1}$, i.e., $\mathbb{P}[R > cMK^{2/3}(\eta\beta d)^{1/3}|E_i] \leq 2^{-c+1}$. Substituting this into equation (25) proves our claim by induction. The desired $O(MK^{2/3}(\eta\beta d)^{1/3}\ln(1/\delta))$ bound on the round complexity is obtained by setting $c \asymp \ln(1/\delta)$ $\square$

