# OpenReview forum: "Diffusion Models are Secretly Exchangeable: Parallelizing DDPMs via Auto Speculation"
_ICML.cc/2025/Conference — ICML 2025 poster_

### Official Review · Reviewer_FuEw · 2025-03-12

**Overall Recommendation:** 3

**Summary:**

This paper proposes and analyzes a parallel sampling scheme for diffusion models. The scheme is simple and natural -- instead of taking a single step from $x_t$ to $x_{t-1}$ during the reverse process in each iteration, multiple steps are taken in parallel from the expected positions $y_s$ given the position $x_t$ at time $t$, as "proposals" for the positions $x_s$, and then, rejection sampling is used to move back in time until a rejected sample if found. The authors show that this scheme gives an $O(d^{2/3})$ time sampling scheme whose performance exactly matches the analogous sequential scheme. Notably, the bound holds without assumption of Lipschitzness of the score, representing the first sublinear in $d$ bound for such distributions, albeit with the use of parallelism. Empirically, the authors show a speed up over DDPM.

**Claims And Evidence:**

Yes, the claims are supported by clear evidence.

**Essential References Not Discussed:**

N/A

**Experimental Designs Or Analyses:**

Yes, they are sound.

**Methods And Evaluation Criteria:**

Yes.

**Other Comments Or Suggestions:**

Generally the paper is well-written, but again, I would have liked to see comparison with prior parallel algorithms.

**Other Strengths And Weaknesses:**

While the scheme is interesting, I would have liked to see experimental comparison with the parallel schemes from prior works -- currently it is unclear how well this scheme performs in practice relative to those approximation schemes. This is perhaps the biggest weakness of this work.

**Questions For Authors:**

Can you perform experiments that compare your scheme to prior parallel sampling schemes?

**Relation To Broader Scientific Literature:**

This paper is part of a line of recent work proposing and analyzing sampling algorithms for diffusion models. It is the first paper that gives a sublinear in $d$ iteration complexity for sampling from distributions without a score Lipschitzness assumption, by making use of parallelism. In contrast, prior work (Gupta et al, 2024; Chen et al, 2024) has shown \emph{polylogarithmic} in $d$ bounds, under a Lipschitzness assumption. Unlike these works which \emph{approximate} the analogous sequential sampling algorithm, this paper proposes a scheme whose samples exactly match the sequential algorithm's.

**Theoretical Claims:**

Yes, they are correct.

---

> ### Author Rebuttal · Authors · 2025-04-01
>
> Thank you for your thoughtful review and for taking the time to engage with our work. We hope to address your concerns below:
>
> ## Comparing ASD with Prior Works
>
> Prior works based on Picard Iterations *can only produce approximate samples* from the DDPM output distribution. In particular, these approaches need to trade-off sample quality against parallel speedup by tuning the error tolerance hyperparameter of the Picard Iteration. On the contrary, ASD is an error-free parallelization scheme that *always produces exact samples* from the DDPM output distribution *for any choice of hyperparameters* (as was also noted by the reviewer). Due to this fundamental difference, we believe the two approaches cannot be compared on equal footing.
>
> Our approach of parallelizing DDPMs via autospeculation represents a novel and fundamentally different "axis of improvement" compared to Picard iterations. In particular, the two approaches are orthogonal to each other and can potentially be combined together for even greater parallel speedups (e.g., by using ASD to sample from each level of the Picard Iteration). This integration presents an exciting direction for future research.
>
> Given these considerations, we believe that the original DDPM sampling method is the most appropriate and fair benchmark for evaluating ASD. We highlight that the primary objective of our experiments was to empirically validate our theoretically guaranteed speedups, while demonstrating a practical application of the hidden exchangeability property of DDPMs (which we believe has potential for broader applications beyond faster DDPM inference).

---

### Official Review · Reviewer_8PeN · 2025-03-13

**Overall Recommendation:** 5

**Summary:**

Diffusion models can be expensive to sample from, since sampling involves integrating a certain stochastic process, and is hence autoregressive (first one needs to take one step, then another step conditional on the result of the previous step, and so on). It would be extremely nice if there were some way to parallelize this sequential process, although it is not obvious whether a way to do this exists.

The authors adapt a recent method for parallelizing autoregressive LLM sampling to diffusion models. They provide a detailed theoretical account of this algorithm and its time complexity as a function of the data distribution's dimensionality, and also illustrate how much it empirically speeds up sampling in a variety of real settings. Their algorithm depends on a relatively surprising feature of diffusion model trajectories that they call "exchangeability".

**Claims And Evidence:**

The authors provide both detailed proofs of their theoretical claims and conduct a variety of helpful experiments to show that their method works well in practice. Both theory and experiment appear to be very high quality.

**Essential References Not Discussed:**

No references come to mind.

**Experimental Designs Or Analyses:**

No, but the authors write clearly and convincingly, so I am reasonably confident their results are correct.

**Methods And Evaluation Criteria:**

Yes. Their algorithm makes sense, and they spend significant real estate to explain its intuition and details. Their experiments also make sense, and they are careful to explain various technical details (e.g., the different relevant senses in which their algorithm can provide a speedup).

**Other Comments Or Suggestions:**

small typo in fig 3 caption, "different sampling method"->"different sampling methods"

**Other Strengths And Weaknesses:**

The paper is well-written and clearly organized. The math is high-quality. The figures look great.

**Questions For Authors:**

My main questions are related to the surprising exchangeability result. The empirical experiments seem to indicate that exchangeability is 'true', or at least close enough in practice to 'true'. But I wonder to what extent the theoretical validity of exchangeability relates to defining the forward process as a pure OU process (Eq. 1). In particular, Eq. 1 does not involve any explicit time-dependence, and seems not to obviously include two popular schemes (VP-SDE, and the VE scheme used by EDM).

First, is it always possible to reparameterize a given forward process (e.g., VE or VP-SDE) to obtain a pure OU process like the one the authors use? This doesn't seem to be true in the case of VE. If it's not true, is it close enough to being true? e.g., one can consider an OU process with $- \epsilon \ \mathbf{x}$ for small $\epsilon > 0$ to model the VE case.

Second, in practice, each forward process generalizes differently (for example, because common discretizations affect the corresponding reverse processes somewhat differently). So they are legitimately different, and not purely reparameterizations of one another. How can the authors' theory account for this? Also, does sampling via the authors' approach affect generalization or sample quality (e.g., FID scores) at all? A theoretical guarantee is one thing, but for various (theoretically interesting!) reasons things may be different in practice.

**Relation To Broader Scientific Literature:**

The authors' proposal makes an interesting bridge between making (sampling from) LLMs more efficient, and making (sampling from) diffusion models more efficient. Their theoretical work builds on a variety of previous formal-math-flavored theory for understanding diffusion models. Their proposal is also related to various other ideas about how to speed up diffusion model inference, and the authors clearly compare their approach to these ideas (e.g., by pointing out that their method has theoretical guarantees related to being error-free, unlike some other methods).

**Theoretical Claims:**

No, but the authors write clearly and convincingly, so I am reasonably confident their results are correct.

---

> ### Author Rebuttal · Authors · 2025-04-01
>
> $\newcommand{\d}{\mathsf{d}}$
> $\newcommand{\vx}{\mathbf{x}}$
> $\newcommand{\vy}{\mathbf{y}}$
> $\newcommand{\vz}{\mathbf{z}}$
>
> Thank you for your thoughtful review and insightful questions which have helped improve our work. We hope to address your concerns below:
>
>
> ## Hidden Exchangeability Beyond OU DDPMs
>
> Thanks for the helpful pointer! **The hidden exchangeability property is not limited to the OU process and actually holds for a large class of generic DDPM formulations, including both VP-SDE and VE-SDE**. This is because both VP-SDE and VE-SDE can be expressed as invertible reparametrizations of the OU process. As a consequence, they are both equivalent to the Stochastic Localization process, and thus, satisfy hidden exchangeability. We shall update our draft to include a proof of this result, which we briefly sketch below:
>
> Consider an arbitrary SDE of the form $\d \vz_t = h(t) \vz_t \d t + \sqrt{g(t)} \d B_t$ where $g, h$ are arbitrary continuously differentiable functions with $g(t) > 0$. Note that $h(t) = -\tfrac{1}{2}g(t)$ recovers VP-SDE and $h(t) = 0$ recovers VE-SDE. Now, consider the OU process $\d \vx_t = - \vx_t \d t + \sqrt{2} \d B_t$ with $\vx_0 = \vz_0$ and let $\vy_t =  m(t,\vy_t) \d t + \d B_t$ denote the SL process as defined in Section 3.1 Eqn 4. Now, define the functions $\alpha(t)$ and $r(t)$ as follows:
> $$ \alpha(t) = \frac{1}{2} \ln \left(1 + \int_{0}^{t} g(s) \exp({-2 \int_{0}^{s} h(u) \d u}) \ \d s\right) $$
> $$ r(t) = \exp(\alpha(t) + \int_{0}^{t} h(s) \d s)$$
>
> Since $g(t) > 0$, $\alpha$ is a strictly increasing function with $\alpha(0) = 0$, i.e., $\alpha$ is an invertible function. Furthermore, $r(t) > 0 $. One can now Apply Ito's Lemma and the time transformation theorem for SDEs (see [1, Thm 8.5.7]) to prove that $\vz_t = r(t) \vx_{\alpha(t)}$, i.e., $\vz_t$ is reparametrizable to the OU process. As discussed in Section 3.1 of our work, $\vy_t = t e^{s(t)} \vx_{s(t)}$ where $s(t) = \tfrac{1}{2} \ln(\tfrac{t+1}{t})$. It follows that $\vz_t$ also maps to the SL process via the following parametrization:
> $$\vy_t = \frac{t e^{s(t)}}{r(\alpha^{-1}(s(t)))} \vz_{\alpha^{-1}(s(t))}$$
> Consequently, **the hidden exchangeability property also applies to $\vz_t$**
>
> ## Effects of Autospeculation on Sample Quality
>
> As elucidated in Tables 1, 2 and 3, our evaluations demonstrate that **ASD consistently achieves the same sample quality as the sequential DDPM implementation** (benchmarked via CLIP and FID scores for Image Generation and Task Success Rates for Robomimic Tasks) These findings corroborate our theoretical guarantee that ASD is an error-free parallelization scheme that always produces exact samples from the original DDPM's output distribution (Theorem 3).
>
> ### References
>
> 1. Oksendal : Stochastic Differential Equations

---

### Official Review · Reviewer_Ed9d · 2025-03-16

**Overall Recommendation:** 2

**Summary:**

This paper reveals the hidden exchangeability inherent in Denoising Diffusion Probabilistic Models (DDPMs) and proposes Autospeculative Decoding (ASD), a novel algorithm that leverages the model itself to generate multi-step speculations and verifies them in parallel. By eliminating auxiliary draft models, ASD achieves a theoretically guaranteed O(K^{1/3}) acceleration over sequential DDPM sampling while preserving zero quality loss. Empirical evaluations demonstrate 1.8-4x practical speedups across image generation (e.g., Stable Diffusion) and robotic control tasks, with CLIP/FID scores and policy success rates on par with original DDPMs.

**Claims And Evidence:**

1.Practical implementations of DDPM typically use discrete steps, whereas the theoretical analysis is based on continuous SDEs. although Theorem 11 analyses the discretisation error, its impact on exchangeability (e.g., whether large step sizes destroy exchangeability) is not explicitly quantified.
2. The effectiveness of ASD relies on exchangeability, and experimental results showing lossless acceleration indirectly support the existence of this property. However, targeted experiments (e.g., distribution consistency tests after replacement increments) were not designed to directly validate exchangeability. Theoretical derivation is rigorous but discretisation effects need further discussion, and experiments indirectly support but lack direct validation.
3. Non-destructiveness not validated in higher dimensional tasks (e.g. video generation) or complex distributions (multimodal). Theoretical proof is sufficient, experimental support is valid but more evidence is needed for scalability.
4. Theorem 4 relies on the covariance decay of the SL process (Theorem 10) and rejection sampling probability bounds, under the assumptions that Tr(Cov[μ])≤βd and step sizes ηk​≤η are reasonable. The derivation adapts the autoregressive framework of Anari et al. (2024a) to continuous spaces. Whether the practical value of β is dimension-independent. If β scales as β=O(1) with respect to d, the theoretical speedup holds. However, if β=O(d), the theoretical guarantees may degrade, weakening the claimed acceleration ratio.

**Essential References Not Discussed:**

The authors correctly cite speculative decoding on arbitrary order autoregressive models. However, a recent extension to continuous spaces is missing:
1. Distilled Decoding 1: One-step Sampling of Image Auto-regressive Models with Flow Matching[J]. arXiv preprint arXiv:2412.17153, 2024.
2. Accelerated Diffusion Models via Speculative Sampling[J]. arXiv preprint arXiv:2501.05370, 2025.

**Experimental Designs Or Analyses:**

In conclusion, the experimental design of the paper is generally reasonable, but there may be room for improvement in statistical significance and detailed description of the experimental setup.
1. The advantages of ASD in terms of error and acceleration ratio compared to Picard's iterative method of Shih et al. (2024) need to be quantitatively compared (e.g., FID vs. speed profiles).
2. ASD's computational cost of rejection sampling (e.g., rejection rate of Algorithm 3) is not quantitatively analysed. High rejection rates may weaken the actual acceleration, and acceptance rate statistics under different tasks need to be added.

**Methods And Evaluation Criteria:**

The self-speculative decoding (ASD) method proposed in the paper addresses the core problem of slow inference of diffusion models, proves the exchangeability of denoised trajectory increments by revealing the equivalence between diffusion models and stochastic localisation (SL), provides a theoretical basis for parallelisation, and is theoretically innovative. With comprehensive assessment criteria: the experimental design of the paper covers multiple dimensions, and the assessment criteria are reasonable and persuasive. The paper has some room for improvement in comparison experiments, but the core validation is still persuasive.

**Other Comments Or Suggestions:**

No.

**Other Strengths And Weaknesses:**

Strengths:
The discovery of hidden exchangeability via the DDPM-SL equivalence is novel, providing a fresh theoretical lens for diffusion models. Extending speculative decoding to continuous spaces without draft models is a significant conceptual leap.
The proposed ASD achieves lossless acceleration (1.8-4x wall-clock speedup) with rigorous guarantees, addressing a critical bottleneck in real-time applications like robotics.
Weaknesses:
The comparative experiments are not comprehensive, and the theorems and explanations are not clear and complete enough. Refer to “Claims And Evidences”, “Theoretical Claims” and “Experimental Designs Or Analyses” sections.

**Questions For Authors:**

1.Practical implementations of DDPM typically use discrete steps, whereas the theoretical analysis is based on continuous SDEs. although Theorem 11 analyses the discretisation error, its impact on exchangeability (e.g., whether large step sizes destroy exchangeability) is not explicitly quantified.
2. The effectiveness of ASD relies on exchangeability, and experimental results showing lossless acceleration indirectly support the existence of this property. However, targeted experiments (e.g., distribution consistency tests after replacement increments) were not designed to directly validate exchangeability. Theoretical derivation is rigorous but discretisation effects need further discussion, and experiments indirectly support but lack direct validation.
3. Non-destructiveness not validated in higher dimensional tasks (e.g. video generation) or complex distributions (multimodal). Theoretical proof is sufficient, experimental support is valid but more evidence is needed for scalability.
4. Theorem 4 relies on the covariance decay of the SL process (Theorem 10) and rejection sampling probability bounds, under the assumptions that Tr(Cov[μ])≤βd and step sizes ηk​≤η are reasonable. The derivation adapts the autoregressive framework of Anari et al. (2024a) to continuous spaces. Whether the practical value of β is dimension-independent. If β scales as β=O(1) with respect to d, the theoretical speedup holds. However, if β=O(d), the theoretical guarantees may degrade, weakening the claimed acceleration ratio.
5. The exchangeability strictly requires equal step sizes ηi​. While the authors mention this condition in Theorem 1, they do not explicitly quantify how unequal ηi​ affects practical ASD performance (tested empirically but not theoretically).
6. In Theorem 11, The proof states E[Σt] is non-increasing in PSD order but does not explicitly cite the Löwner-Heinz theorem, which is required for this claim.
7. The advantages of ASD in terms of error and acceleration ratio compared to Picard's iterative method of Shih et al. (2024) need to be quantitatively compared (e.g., FID vs. speed profiles).
8. ASD's computational cost of rejection sampling (e.g., rejection rate of Algorithm 3) is not quantitatively analysed. High rejection rates may weaken the actual acceleration, and acceptance rate statistics under different tasks need to be added.

**Relation To Broader Scientific Literature:**

1.While prior diffusion acceleration methods (e.g., DDIM, DPM-Solver) trade quality for speed via deterministic approximations or reduced steps, ASD achieves lossless acceleration through parallelism, akin to Shih et al. (2024)’s parallel sampling but with theoretical guarantees. Unlike recent parallel DDPM methods (Gupta et al., 2024; Chen et al., 2024) that require restrictive Lipschitz assumptions, ASD operates under minimal second-moment conditions, broadening applicability.
2.The work adapts speculative decoding (Leviathan et al., 2023; Chen et al., 2023), originally designed for discrete autoregressive models (e.g., LLMs), to continuous-state diffusion models. Crucially, it eliminates the need for a draft model—a limitation in prior speculative methods—by exploiting the exchangeability property. This aligns with Anari et al. (2024a)’s framework for any-order autoregressive models but addresses the unique challenges of infinite token spaces (i.e., continuous domains).

**Theoretical Claims:**

1.The exchangeability strictly requires equal step sizes ηi​. While the authors mention this condition in Theorem 1, they do not explicitly quantify how unequal ηi​ affects practical ASD performance (tested empirically but not theoretically).
2. In Theorem 11, The proof states E[Σt] is non-increasing in PSD order but does not explicitly cite the Löwner-Heinz theorem, which is required for this claim.

---

> ### Author Rebuttal · Authors · 2025-04-01
>
> $\newcommand{\bE}{\mathbb{E}}$
> Thank you for your review and for taking the time to engage with our work. We hope to address your concerns below:
>
> ## Analysis of Discrete Steps
>
> We highlight that **all our theoretical guarantees for ASD directly consider the discrete-time regime**. To this end, our proof of Theorem 2 on the correctness of ASD *does not* make any use of the continuous time SL process, and relies only on the properties of the Gaussian Rejection Sampler. Our proof of Theorem 3, which analyzes the discrete-time parallel complexity of ASD, uses the SL process *only as an analytic tool* to interpolate between the ASD increments. Such interpolation arguments are ubiquitous in the sampling literature, particularly in the analysis of diffusion models [1,2] as well as gradient-based sampling algorithms like Langevin Monte Carlo. [3,4]
>
> ## Impact of Discretization on Exchangeability
>
> While our discussion on hidden exchangeability in Section 3.1 focuses on the continuous case for ease of exposition (to provide intuition behind why the arguments of Anari et. al.'24 can be extended to diffusion models), **the impact of discretization on hidden exchangeability is appropriately quantified in the proof of Theorem 3**. In particular, the proof of Theorem 3 establishes that, **as long as the step-size is not too large, ASD increments aproximately satisfy the hidden exchangeability property**. This holds because the discrete trajectory of ASD is statistically indistinguishable from that of the continuous SL process (which exactly satisfies hidden exchangeability as per Theorem 1). We rigorously quantify this in the proof of Theorem 16 in Equations 20 and 21 by proving that the TV distance between ASD and the continuous process is $O(\sqrt{\eta \beta d})$. Hence, the two can be made statistically indistinguishable by choosing $\eta \asymp \tfrac{1}{\beta d}$.  We shall update our draft to highlight this point in the main text.
>
> ## High Dimensional Tasks
>
> Our experiments on Pixel Diffusion demonstrate the effectiveness of ASD on a 196,608 dimensional task. Evaluations on more complex modalities such as video is an interesting avenue of future work which we are unable to pursue at present due to resource constraints.
>
> ## Unequal Step-Sizes
>
> We respectfully disagree with the claim that the case of unequal step-sizes is not analyzed theoretically. We highlight that **Theorem 1 proves a general time invariance property** for the distribution of stochastic localization increments, **which holds even for unequal step-sizes $\eta_i$**. We call this time invariance the hidden exchangeability property as it reduces to exchangeability of increments when the step-sizes are equal. However, the key result of Theorem 1 is agnostic to the choice of step-sizes. Consequently, **all our theoretical guarantees for ASD are directly applicable to unequal step-sizes.**
>
> ## Assumption on $\beta$
>
> We first note that $\beta = O(1)$ is a standard assumption in the theory of diffusion models [1,2]. Secondly, for arbitrary $\beta$, the canonical choice of step-size is $\eta \asymp \tfrac{1}{\beta d}$ (e.g. [1] uses $\eta \asymp \tfrac{1}{M_2}$ where $M_2 = \beta d$, see also [2, Appendix C]). Under this setting, the parallel runtime of ASD as per Theorem 4 is $O(K^{2/3})$. Hence, **the $K^{1/3}$ parallel speedup guarantee continues to hold.**
>
> ## References
>
> Thank you for the helpful references, which we weren't aware of at the time of writing. We shall update our draft to include them.
>
> ## Löwner-Heinz Theorem
>
> $\bE[\Sigma_t]$ being a decreasing function in the Löwner order directly follows from the facts that: 1. $\tfrac{d{\bE[\Sigma_t]}}{dt} = -\bE[\Sigma_t^2]$ and, 2. $\bE[\Sigma_t^2]$ is a PSD matrix. To our understanding, the Löwner-Heinz theorem (which deals with the operator monotonicity of $t^p$ for $p \in [-1,1]$) is not relevant here. Please let us know if we have made a mistake.
>
> ## Comparison to Shih et. al.
>
> Please refer to our response to Reviewer FuEw for a detailed discussion on why our results and that of Shih. et. al. cannot be compared on equal footing (and why our work and Shih. et. al. represent orthogonal axes of improvement for fast DDPM inference)
>
> ## Quantifying the Acceptance Rates
>
> We highlight that the acceptance rate is quantitatively analyzed in the proof of Theorem 4. In particular,  the total probability of rejection is upper bounded by $O(\sqrt{K \theta \eta \beta d})$
>
>
> ### Refs
>
> 1. Chen et. al. : Sampling is as easy as learning the score
> 2. Benton et. al. : Convergence Bounds for DDPMs via Stochastic Localization
> 3. Vempala & Wibisono : Rapid Convergence of the Unadjusted Langevin Algorithm
> 4. Balasubramanian et. al. : Towards a Theory of Non-Log Concave Sampling
> 5. Anari et. al. : Parallel Sampling via Counting

---

### Decision · Program_Chairs · 2025-05-01

**Decision:**

Accept (poster)

**Comment:**

The authors adapt the idea of speculative decoding in LLMs to provide a technique that uses parallelization to accelerate the process of drawing samples from a denoting diffusion model (DDPM).  Their innovation is based on an exchangeability property in the increments of the denoising process.  This provides exact sampling, in the sense that the distribution of the DDPM is not altered by the parallelization, but reduces iteration complexity from O(d) to O(d^{2.3}), where d is the dimension of the samples.

This seems like a very interesting idea, on a computational problem that is quite meaningful and potentially impactful. Overall, reviewers were positive, with some concerns that certain aspects could be made clearer -- more information on the exchangeability result, for example, which underpins the method; and potential "non-ideal" aspects of practical sampling processes, such as discretization effects, could undercut the theory in practice (although the authors argue that the samples remain of good quality). Reviewer FuEw also points out a number of missing references on parallel sampling using additional assumptions and providing approximate samples; comparing to these methods would strengthen the paper.